# Machine learning–based observation-constrained projections reveal elevated global socioeconomic risks from wildfire

Yan Yu [1,6], Jiafu Mao [2,6 ✉], Stan D. Wullschleger[2], Anping Chen [3], Xiaoying Shi[2], Yaoping Wang [4], Forrest M. Hoffman[5], Yulong Zhang[4] & Eric Pierce [2]

Reliable projections of wildfire and associated socioeconomic risks are crucial for the development of efficient and effective adaptation and mitigation strategies. The lack of or limited observational constraints for modeling outputs impairs the credibility of wildfire projections. Here, we present a machine learning framework to constrain the future fire carbon emissions simulated by 13 Earth system models from the Coupled Model Intercomparison Project phase 6 (CMIP6), using historical, observed joint states of fire-relevant variables. During the twenty-first century, the observation-constrained ensemble indicates a weaker increase in global fire carbon emissions but higher increase in global wildfire exposure in population, gross domestic production, and agricultural area, compared with the default ensemble. Such elevated socioeconomic risks are primarily caused by the compound regional enhancement of future wildfire activity and socioeconomic development in the western and central African countries, necessitating an emergent strategic preparedness to wildfires in these countries.

[1] Department of Atmospheric and Oceanic Sciences, School of Physics, Peking University, Beijing, China. [2] Environmental Sciences Division and Climate Change Science Institute, Oak Ridge National Laboratory, Oak Ridge, TN, USA. [3] Department of Biology and Graduate Degree Program in Ecology, Colorado State University, Fort Collins, CO, USA. [4] Institute for a Secure & Sustainable Environment, University of Tennessee, Knoxville, TN, USA. [5] Computational Sciences and Engineering Division and Climate Change Science Institute, Oak Ridge National Laboratory, Oak Ridge, TN, USA. [6] These authors contributed equally: Yan Yu, Jiafu Mao. ✉email: maoj@ornl.gov

Wildfires represent a major ecosystem disturbance and aerosol emission source, affecting the global carbon budget, climate, and human life[1,2]. A recent surge of disastrous fires caused enormous social disruptions and huge economic losses[3]. During the 2019–2020 Australian bushfire season, a series of major wildfires burned more than 190,000 km[2], costing more than 20 billion 2020 USD, and killing at least 33 people[4]. Through global climate change[5], human influence on fire ignition[6], land-use/land-cover change[7], and complex response of the land biosphere to human-induced climate change and $CO_2$ fertilization[8], anthropogenic activity has remarkably altered wildfire behavior and its environmental risks[9–11] at various temporal and spatial scales. These scale-dependent human-fire feedbacks also complicate the future projection of wildfire regimes across the globe (e.g., size, frequency, and intensity, and their socioeconomic impacts). Nonetheless, accurate spatio-temporal wildfire prediction is essential to the estimate of future socioeconomic risks because of the tight linkage between wildfire regimes and fire-relevant socioeconomic effects, and the highly heterogeneous nature of the projected socioeconomic development[3].

Process-based Earth system approaches, such as the use of Earth system models (ESMs), have the potential to account for many human-vegetation-fire-climate interactions and are thus suggested as a practical way to predict future changes of wildfire and associated socioeconomic exposure (e.g., population, gross domestic product [GDP], and agricultural area). Notwithstanding, reliable long-term wildfire projections remain highly uncertain and challenging, primarily because even the state-of-the-science ESMs are still limited in characterizing the human-vegetation-fire-climate feedback[12]. Such uncertainties potentially lead to biases in the simulated historical fire carbon emission[13] by ESMs participating in the latest Coupled Model Intercomparison Projection phase 6 (CMIP6)[14] (Supplementary Fig. 1), casting doubt on the credibility of the projected wildfire evolution from the default models. Given the uncertainty in the dynamically simulated wildfire[15,16], previous efforts on wildfire projections were often focused on the use of fire weather simulated by ESMs or global climate models as an emulator for fire potential[17,18]. However, the linkages between fire weather and wildfire activity are greatly affected by other factors, including terrain, fuel abundance, fuel moisture content, source of ignition, and their interactions[19–22]. Although current ESMs only include incomplete and highly parameterized driving processes for fire (Supplementary Table 1), the involved physics-based coupling among fire, climate, ecosystem, and human activities across different scales (e.g., consistent mechanistic relationships between variability in fire-relevant variables, such as air temperature, precipitation, and vegetation coverage, and persistent sensitivity of these climate or ecosystem variables to external anthropogenic forcings) sets the basis for linking future fires with historical states of these components in the ESMs[23,24]. Benefiting from both the ESM-simulated history-future relationships and the availability of multitype fire-relevant historical observations (e.g., fire carbon emission, air temperature, precipitation, and leaf area index), we hypothesize that constraining the ESM wildfire estimates by observations is a potentially valid approach for reducing spatial inaccuracies in global wildfire projections and related socioeconomic risks.

Effective observational constraint of future wildfire changes takes advantage of methodologies developed for projecting other Earth system features (e.g., tropical land carbon[25] and ecosystem photosynthesis[26]) but needs to address additional challenges. The emergent constraint (EC) approach has demonstrated robust capability in reducing the uncertainty in characterizing or projecting Earth system variables simulated by a multimodel ensemble[25,26]. The concept of EC relies on the existence of a tight regression across ESMs between a quantity of interest that is difficult or impossible to measure (e.g., a future state) and a second, measurable variable[27]. Therefore, successful EC applications require a sufficient number of models with diverse structures and parameters. However, this is not applicable to wildfire projections using the latest ESMs participating in CMIP6. Indeed, at the time of this analysis, only 13 of the currently available CMIP6 models provide fire carbon emissions in both historical and future simulations (Supplementary Table 1), which is insufficient to perform a traditional EC analysis. Furthermore, traditional EC implementations only establish linear relationships between a limited number of constraining factors and projected variables, but wildfire-induced carbon emissions can result from complex, nonlinear integration of meteorological, ecological, and socioeconomic states[3], causing largely insignificant linear relationships between future fire carbon emissions and historical constraining variables across the analyzed ESMs, especially over the currently fire-prone regions (Supplementary Fig. 2). Furthermore, whereas the traditional EC has been successfully applied to large-scale averaged quantities (e.g., tropical land carbon[25] and global terrestrial photosynthesis[26]), this analytical framework may not be suitable for projections of variables of local interest, such as wildfire regimes (Supplementary Fig. 2), whose detailed spatial structures are critically needed for estimating their socioeconomic impacts. Other studies have applied performance-based approaches such as bias-correction[28] and model-weighting[29], or process-oriented methods such as multiple diagnostic ensemble regression[30] for reducing the uncertainty in multimodel projections of various Earth system variables. However, these constraining efforts normally rely on univariate and temporally stable assumptions about model performance and ambiguous, linear relationships of the observed metrics and future projections[31]. Therefore, the observation-constrained projection of global wildfire requires more advanced methodologies that can both vigorously capture complex, cross-sector interactions with limited samples of ESM ensembles and accurately resolve the detailed structure of wildfire regimes to better inform future socioeconomic exposure to fire.

This study develops a machine learning–based analytical framework (Supplementary Fig. 3) to establish an observation-constrained projection of global fire carbon emissions and socioeconomic risks using 38 members from 13 CMIP6 ESMs and various sources of observations. These ESMs provide coupled carbon-ecosystem-climate simulations with a wide range of processes and parameterizations included[14]. Their terrestrial components typically contain fire models with process-based and/or data-based parameterization for various landscapes, accounting for effects of changes in both land surface meteorological states, vegetation-soil conditions and human activities on fire regimes (Supplementary Table 1 and reference therein). Inspired by the EC concept and motivated by the complex influencing factors of wildfire, machine learning techniques (MLT) are implemented to quantify the emergent relationships between projected global fire carbon emissions and historical, observed climate, terrestrial ecosystem, and socioeconomic states, using the complete historical-future spatial patterns simulated by the ESMs (see Methods section). MLT provide useful tools to investigate the nonlinear and complex effects of natural and anthropogenic factors on wildfire activities[32–34], and have been successfully used for predicting seasonal fire carbon emissions and burned areas in Africa[35]. Here, multiple MLT are first trained to capture the ESM-simulated relationships among the selected historical climate, terrestrial ecosystem, and socioeconomic variables and future multi-decadal wildfire-induced carbon emissions under the Shared Socioeconomic Pathway (SSP) 5-85[36]. Observed, historical

environmental (e.g., fire carbon emission, leaf area index [LAI], soil moisture, temperature, precipitation, wind, relative humidity, flash rate, and orography) and socioeconomic (e.g., land use and population) variables (Supplementary Table 2 and reference therein) are subsequently fed into the trained MLT models, resulting in a multimodel, multi-data set ensemble of observation-constrained projections of future global distribution of fire carbon emissions for each decade. These driving variables of wildfires are selected so that their nonlinear combinations as determined by MLT reflect the fuel abundance[8] (LAI, temperature, precipitation), fuel moisture[37] (soil moisture, relative humidity, precipitation, temperature), fire spread conditions[38] (wind and orography), and ignition sources[39] (flash rate, land use and population). The socioeconomic risks associated with future wildfires are then quantified using the default and observation-constrained ensemble projections of fire carbon emissions, along with the population, GDP, and agricultural area projected under the SSP5-85[36] (see Methods).

Here we apply the MLT-based analytical framework to observationally constrain fire carbon emissions and their socioeconomic risks projected by CMIP6. This approach leads to a weaker increase in global fire carbon emissions but higher increase in global wildfire exposure in population, gross domestic production, and agricultural area during the twenty-first century, compared with the default ensemble.

## Results

### Evaluating observation-constrained fire carbon emissions for the historical period.
The MLT-based observational constraint substantially boosts the consistency between simulated and observed wildfire activities for the validation period, 2007–2016, in terms of both magnitude and spatial pattern of global fire carbon emissions (Figs. 1 and 2). The observation-constrained product substantially reduces the overestimation of fire carbon emissions over sparsely vegetated regions (mainly for EC-Earth3 models, Supplementary Fig. 1), tropical rainforests (mainly for EC-Earth3 models), northern boreal regions (mainly for MRI-ESM2.0), and densely populated regions in North America and Europe (mainly for CNRM-ESM2.1 and MPI-ESM1.2-LR), as well as the underestimation of fire carbon emissions over the savannahs in Africa from most analyzed ESMs (Fig. 1). Relatively large error between the observation-constrained and observed fire carbon emissions remains in the present fire-prone regions (e.g., tropical and subtropical Africa, subtropical South America, and southeast Asia) (Fig. 1c). During the validation period, the root mean square error (RMSE) between the simulated multimodel mean and observed historical annual total fire carbon emissions decreases from 0.020 to 0.014 (0.010–0.017, 10th–90th percentile across ensemble members) kg m$^{-2}$ yr$^{-1}$, and the squared spatial correlation ($R^2$) between the observed and simulated multimodel mean, decadal averaged fire carbon emissions increases from 0.36 to 0.66 (0.47–0.92) (all $p$s < 0.001). The error metrics, namely RMSE and $R^2$, produced by the MLT-based observation constraint are significantly better than those derived from the traditional EC approach (Fig. 2), demonstrating the effectiveness of the current approach in resolving the spatial distribution of historical wildfires. The observation constraint also produces a more realistic estimation of historical global fire carbon emissions (Fig. 3a) and their socioeconomic risks compared with the default ensemble (Fig. 4a, d, g). Furthermore, individual ESMs exhibit improved spatial consistency with the observational constraint, with a reduction in RMSE by 46% (NorESM2-LM) to 74% (MRI-ESM2.0) across models and an increment in R$^2$ by 0.30 (E3SM-1.1) to 0.56 (EC-Earth3-CC) (Fig. 2). Such vast improvements in reproducing the historical intensity and spatial pattern of global

fire carbon emissions among all the analyzed ESMs demonstrate the advantage of the MLT-based observational constraint in enhancing the reliability and confidence of the resultant global wildfire projections.

### Model-projected global fire carbon emissions in the twenty-first century.
Compared with the original, unconstrained multimodel ensemble, the MLT-based observational constraint leads to a reduced magnitude, a less pronounced future increase, and a much narrower spread of global fire carbon emissions (Fig. 3). The default multimodel ensemble projects a 6.0% (0.6%–9.4%) decade$^{-1}$ increase in global total fire carbon emission from $2.7 \times 10^3$ ($1.6 \times 10^3$ – $4.7 \times 10^3$) Tg yr$^{-1}$ during the 2010s to $4.0 \times 10^3$ ($2.1 \times 10^3$ – $1.4 \times 10^4$) Tg yr$^{-1}$ during the 2090 s (Fig. 2a). According to the observation-constrained multimodel ensemble, the global total fire carbon emission is projected to increase by 4.1% (2.6% – 7.2%) decade$^{-1}$ from $2.0 \times 10^3$ ($1.7 \times 10^3$ – $2.4 \times 10^3$) Tg y$^{-1}$ during the 2010s ($2.0 \times 10^3$ Tg y$^{-1}$ during the 2010s reported by two observational data sets) to $2.8 \times 10^3$ ($2.7 \times 10^3$ – $3.4 \times 10^3$) Tg y$^{-1}$ during the 2090 s. The global fire carbon emission is projected to increase monotonically during the twenty-first century, indicated by the original multimodel mean and 9 out of the 13 analyzed ESMs. However, four ESMs (CESM2, CESM2-WACCM, NorESM2-LM, and NorESM2-MM) that share the same land component, namely the Community Land Model (CLM) version 5 (Supplementary Table 1), simulate a reduced global fire carbon emission during the first half of the twenty-first century, resulting in a relatively stable fire carbon emissions from 2010s to 2050s produced by the observation-constrained ensemble (Fig. 3a). An exclusion of the NorESM2 models leads to a slightly elevated future increase in fire carbon emissions produced by the observational constraint, especially over the northern extratropical land surface (Supplementary Fig. 4).

The projected evolution of fire carbon emission distribution is substantially altered by the observational constraint (Fig. 3b–d and Supplementary Fig. 5). Overall, the observation-constrained ensemble estimates a robust increase in future fire carbon emissions over most of the global land, with only a few regions showing insignificant decrease (i.e., the northern boreal region in Eurasia and the North American Great Lakes region) (Fig. 3c). In contrast, the default multimodel ensemble consistently projects increased fire carbon emissions over the Northern Hemispheric subtropical and boreal regions, as well as southern Africa and southern Australia, whereas the assessed ESMs produce divergent future changes in fire carbon emissions for most of South America, tropical Africa, southern Asia, and northern Australia (Fig. 3b and Supplementary Fig. 5).

Specifically, the default and observation-constrained ensembles demonstrate a consistent, positive future trend in fire carbon emissions mainly for tropical forests in the Maritime Continent, Central America, and Amazon, the tropical and subtropical savannahs in northeastern South America, southern and eastern sub-Saharan Africa, and southern Australia, as well as mid-latitude grasslands and croplands in eastern Europe and central and eastern Asia. Across the tropical forests and savannahs in West Africa, Congo, northern Australia, and eastern South America, the default multimodel ensemble (particular ESMs with CLM as the land component, Supplementary Fig. 6 and Supplementary Table 1) projects a future reduction in fire carbon emissions, although the observation-constrained ensemble indicates a significantly positive trend. The original multimodel ensemble (mainly from CNRM-ESM2.1, Supplementary Fig. 6) also simulates enhanced fire carbon emissions across the subtropical, temperate, and boreal forests in North America,

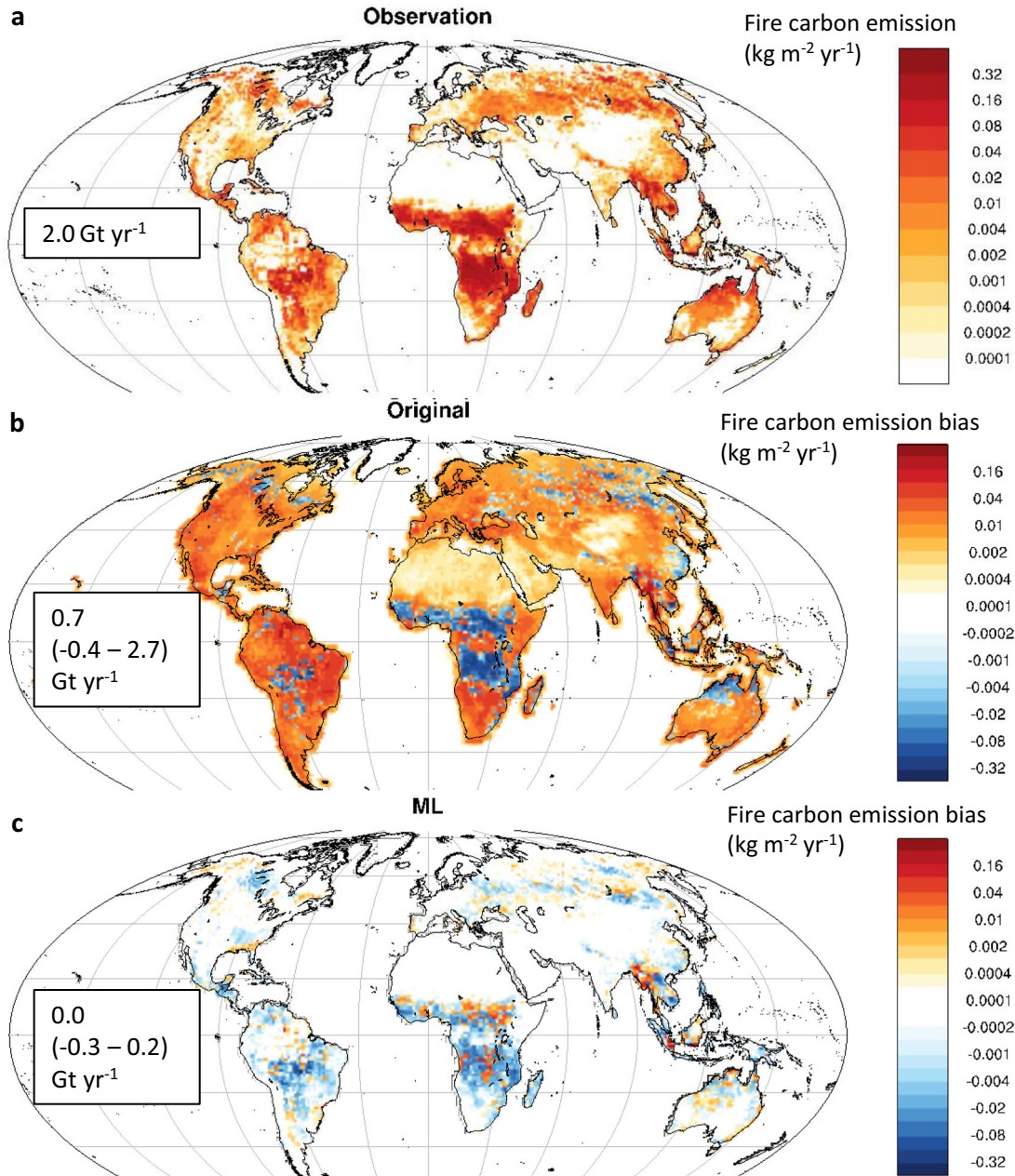

**Fig. 1 Historical fire carbon emissions during 2007–2016 simulated by 13 Earth system models (ESMs) without and with the observational constraint.**
**a** Observed mean fire carbon emissions (kg m$^{-2}$ yr$^{-1}$), averaged across two observational data sets. The global total fire carbon emission and its uncertainty range is marked. **b** Spatial distribution of the bias in original, unconstrained multimodel mean fire carbon emissions (kg m$^{-2}$ yr$^{-1}$). **c** Spatial distribution of the bias in observation-constrained multimodel, multi–data set mean fire carbon emissions (kg m$^{-2}$ yr$^{-1}$). The bias of global total fire carbon emission and its uncertainty range (10th–90th percentiles) is marked in the corresponding panels **b**, **c**.

Europe, and east Asia during the twenty-first century (Fig. 3b) to a larger extent than the observation-constrained multimodel ensemble (Fig. 3c), and the observation-constrained ensemble estimates a more drastic increase in fire carbon emissions in the northeast United States and the Appalachian Mountains (Fig. 3c). In the sparsely vegetated north Africa, Middle East, and central Asia, the default multimodel ensemble simulates an increase in fire carbon emissions (Fig. 3b), mainly contributed by the EC-Earth3 and GFDL-ESM4, corresponding to the bias in their simulated historical fire carbon emissions (Supplementary Figs. 1 and 6); however, these regions show minimal future changes in fire carbon emissions in the observation-constrained ensemble. Such inconsistency in the spatial distribution of projected fire

carbon emissions trend between the constrained and unconstrained multimodel ensembles leads to their distinct projected evolution of the latitudinal fire carbon emissions (Fig. 3d). Because of the projected future inhibition of fire carbon emissions from eastern South America, Congo, south Asia, and northern Australia, the default multimodel ensemble estimates a weakened increase in fire carbon emission from 10°S–Equator and Equator–10°N by 0.5% (−1.6%–1.8%, 10th–90th percentiles among models) decade$^{-1}$ and 0.4% (−1.7%–1.1%) decade$^{-1}$ during the twenty-first century, roughly a quarter of the trend from the observation-constrained ensemble. On the contrary, for the Great Lakes region and boreal Eurasia where the default ensemble projects a decrease in fire carbon emissions, the

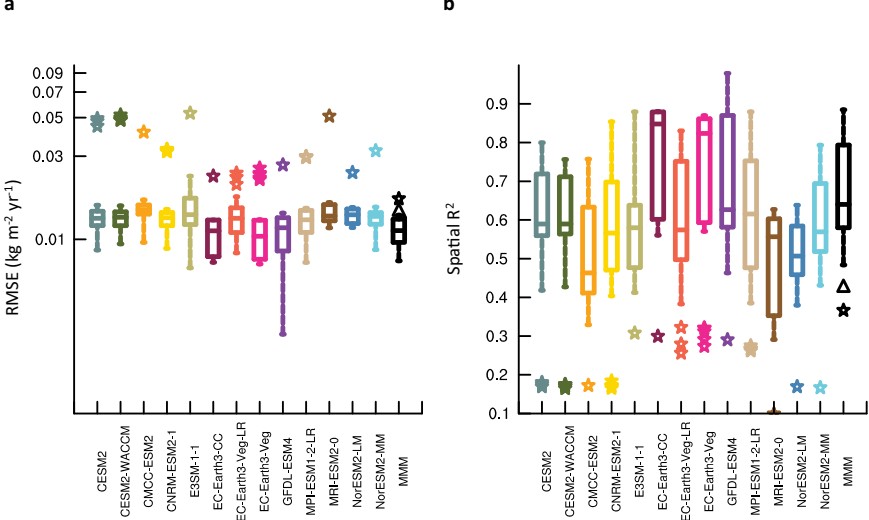

**Fig. 2 Accuracy of historical fire carbon emissions during 2007–2016 simulated by 13 Earth system models (ESMs) without and with the observational constraint. a** Root mean square error (RSME) (kg m$^{-2}$ yr$^{-1}$) between the observed, decadal mean global fire carbon emission and the original (star representing each ensemble member) and observation-constrained (boxplot, representing the 10th, 25th, 50th, 75th, and 90th percentiles in the multi–data set ensemble) simulations from 13 ESMs and their multimodel mean. **b** Squared spatial correlation ($R^2$) between the observed, decadal mean global fire carbon emission and the original (stars) and observation-constrained (boxplot) simulations from each of the 13 ESMs and their multimodel mean. The black triangles in a and b indicate the RMSE and $R^2$ produced by the traditional emergent constraint (EC) approach that constrains fire carbon emissions during 2007–2016 with fire carbon emissions during 1997–2006.

observation-constrained ensemble predicts an increase in fire carbon emissions by 0.6% (0.5%–0.7%) decade$^{-1}$, 0.5% (0.2%–0.9%) decade$^{-1}$, and 0.03% (−0.1%–0.6%) decade$^{-1}$ from 40°N–50°N, 50°N–60°N, and 60°N–70°N, respectively, with much weaker trends than the default ensemble.

**Model-projected socioeconomic risks from wildfire in the twenty-first century.** The distinct future evolutions of fire carbon emissions intensities and spatial patterns projected by the original and observation-constrained multimodel ensembles also lead to their divergent projections on socioeconomic exposures to wildfires at both global and national scales (Fig. 4 and Supplementary Table 3). Under the projected changes in global fire carbon emissions and socioeconomic development, the global wildfire exposure in population, GDP, and agricultural area is projected to increase by 5.5% (5.0%–6.2%) decade$^{-1}$, 40.6% (33.7%–48.5%) decade$^{-1}$, and 2.5% (1.9%–3.7%) decade$^{-1}$, respectively, during the twenty-first century, based on the observation-constrained multimodel, multi–data set ensemble average (Fig. 4a, d, g). As a comparison, the original multimodel ensemble projects similar absolute changes as the observation-constrained ensemble, yet with a weaker relative increase by 3.2% (1.1%–7.9%), 12.6% (7.0%–28.5%), and 1.8% (0.9%–5.5%) decade$^{-1}$ in the global wildfire exposure in population, GDP, and agricultural area, respectively (Fig. 4a, d, g) because the models simulate higher historical wildfire risks. The further intensified socioeconomic risks from wildfire estimated by the observation-constrained versus the default ensembles are primarily owing to the in-phase enhancement of future wildfire activities and expected rapid socioeconomic development in the currently fire-prone west and central African countries (Fig. 3c), in direct opposition to the projected reduction in wildfires over these regions by the default ensemble (Fig. 3b). As a result, the observation-constrained ensemble indicates elevated socioeconomic risks from wildfires in the west and central African countries to a larger extent than the default ensemble during the twenty-first century (Fig. 4b, c, e, f, h, i). Notably, although both the default and observation-constrained ensembles highlight African countries in their list of top 10 countries facing the greatest

relative changes in socioeconomic risks from wildfires during the twenty-first century, the observation-constrained ensemble particularly tags west African countries, such as Niger and Sierra Leone, as the most vulnerable countries in all the three metrics of socioeconomic risks from future wildfires (Supplementary Table 3).

## Discussion

Our MLT-based analytical framework integrates the physical processes contained in the ESMs and an observational constraint to reduce the uncertainty in ESM simulations of historical and future fire carbon emissions. ESMs include self-consistent dynamical interactions among fire carbon emission, climate, and ecosystem, which provide the basis for connecting future states with historical, observable states. As characterized by the fluctuation-dissipation theorem, there are profound mechanistic relationships between short-term spatiotemporal variability in near-linear systems or linear approximations of more complex systems (e.g., the climate and ecosystem) and their long-term response to external forcings (e.g., anthropogenic forcings)[23,24]. This physics-based historical-future connection facilitates the application of the EC concept in making long-term projections of the climate and ecosystem, as well as the resultant global fire carbon emissions and socioeconomic risks. However, the traditional EC framework appears less reliable in the projection of fire carbon emissions (Supplementary Fig. 2), likely attributed to the following factors[31,40,41]: (1) complex interactions among fire, climate, ecosystem, and socioeconomic variables, inadequately captured by the linear assumption adopted in the traditional EC framework; and (2) the relatively small collection of fire simulations and limited diversity in fire parameterizations for the available ESMs (Supplementary Table 1), providing insufficient sample for applying the traditional EC framework. Building on the EC concept, our MLT-based observational constraining framework advances further in several key perspectives: (1) This framework integrates the information from multitype observations of fire-relevant variables with the mechanistic history-future relationship encompassed in ESMs. (2) This framework relies on the power of MLT in capturing nonlinear and interactive

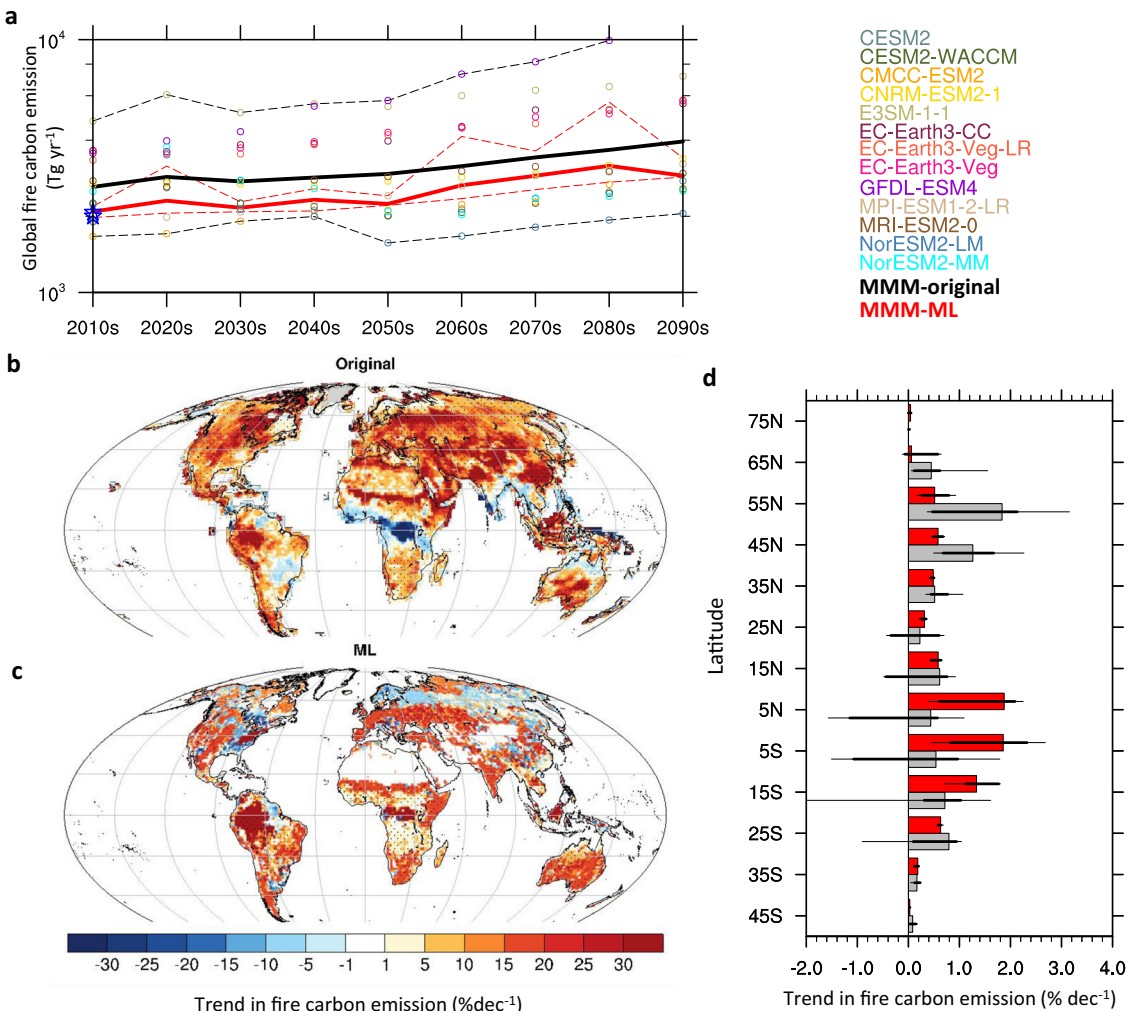

**Fig. 3 Global and latitudinal evolution of fire carbon emissions from the original and observation-constrained multimodel ensembles. a** Time series of global total fire carbon emission ($Tg\,y^{-1}$) from the 2010s to 2090s, according to the original individual Earth system models (ESMs) (circles), their multimodel mean (thick black line), and the observation-constrained multimodel, multi–data set mean (thick red line). The pink shading with dashed boundaries indicates the 10th and 90th percentiles in the multimodel, multi–data set observation-constrained ensemble; and the gray shading with dashed boundaries indicates the 10th and 90th percentiles in the original multimodel ensemble. The blue stars indicate the observed global fire carbon emission from two data sets. **b** Unconstrained and **c** constrained multimodel mean trend in fire carbon emissions (units: change per decade as the percentage of the historical fire carbon emissions in the 2010s) from the 2010s to 2090s. Stitches indicate areas with a robust trend in fire carbon emissions, with a consistent sign of trend among at least 80% of the ensemble members. **d** Trend in the total fire carbon emission (% of the historical fire carbon emission in the 2010s per decade) from each 10° zonal band during the 2010s to 2090s, according to the original (multimodel mean: gray bars; 25th–75th percentiles: thick horizontal lines; 10th–90th percentiles: thin horizontal lines) and observation-constrained (multimodel mean: red bars; 25th–75th percentiles: thick horizontal lines; 10th–90th percentiles: thin horizontal lines) ESM simulations.

relationships between the simulated historical joint states in fire-climate-ecosystem-socioeconomics and future wildfire activities. (3) By taking advantage of the complete spatial pattern provided by the relatively small collection of ESMs, this framework expands the sample size of the training data, thereby enhancing MLT model fitting efficiency and resolving the desired spatial pattern of future wildfire regimes. Benefiting from the inclusion of the complete spatial sample, this observational constraint leads to a consistent and substantial error reduction in simulated global wildfire distribution (Figs. 1 and 2), demonstrating the robustness of our analytical framework even with just 13 ESM ensembles. Indeed, our MLT-based framework also shows satisfactory efficiency in error reduction with only 6 CMIP6 ESMs that simulate burned area fractions (Supplementary Fig. 7). Although the current MLT-based EC framework improves the spatial accuracy of original ESM-simulated fire carbon emissions during the historical validation period, the performance of our framework in the future decades

partially relies on the accuracy of ESMs' physical processes (e.g., complex responses of fire regimes to various natural and anthropogenic forcings) and must be further evaluated. Not surprisingly, the performance of our MLT-based observational constraint largely relies on the spatial resolution of the input ESM data (Supplementary Fig. 8) because finer resolution leads to both expanded training data sample and resolved spatial patterns. Such detailed structure of the target variable as projected by our MLT-based observational constraining framework facilitates accurate assessment of socioeconomic risks in the historical validation period (Fig. 4) and potentially improved future projections, leading to strategic implications for local and regional stakeholders. Similar frameworks can benefit from the projection of other climate and ecological variables that are of local interest (e.g., drought, heatwave, flooding, primary productivity) and their socioeconomic influences.

The MLT-based observational constraint modifies the intensity, distribution, and future projections of global wildfire carbon

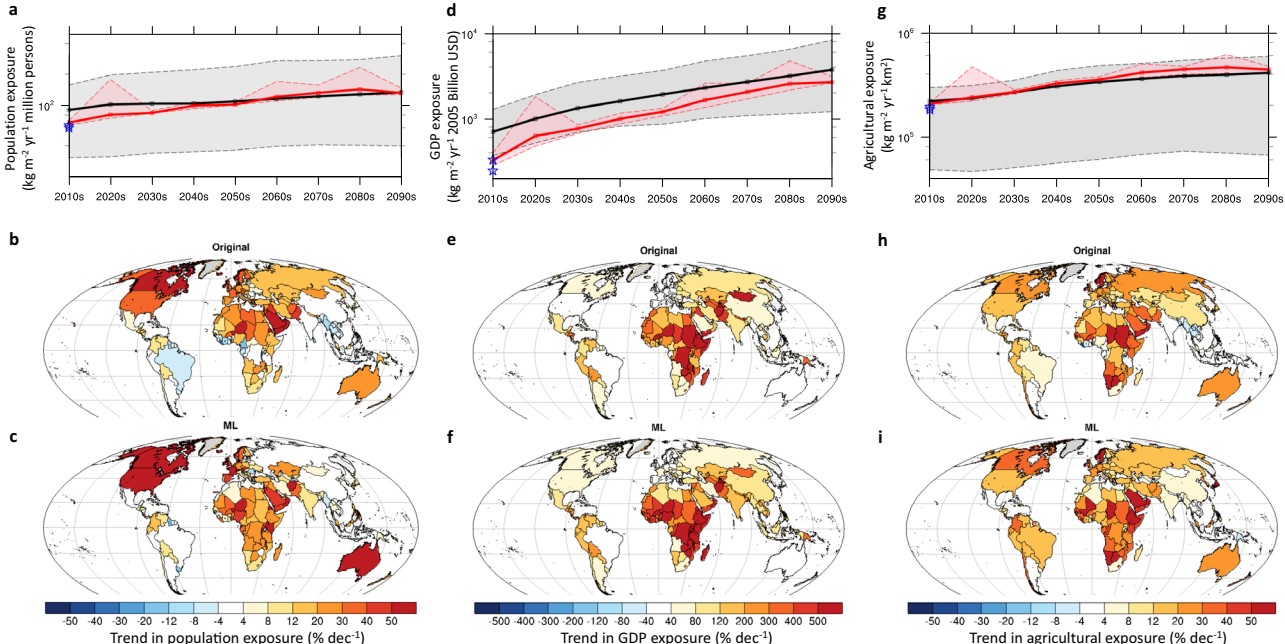

**Fig. 4 Global and national evolution of socioeconomic risks from wildfire carbon emission during the 2010s to 2090 s. a** Original (black line with gray shading) and observation-constrained (red line with pink shading) time series of global total exposure to fire carbon emission in **a**. population (kg m$^{-2}$ yr$^{-1}$ million person), **d** gross domestic product (GDP) (kg m$^{-2}$ purchasing power parity 2005 billion USD y$^{-1}$), and **g** agricultural area (kg m$^{-2}$ km$^2$ y$^{-1}$). The blue stars indicate the observed exposure to fire carbon emission in population, GDP, and agricultural area, respectively, during the 2010s. The lines represent the multimodel ensemble mean. The shadings represent the 10th and 90th percentiles among ensemble members. **b**, **e**, **h** original and **c**, **f**, **i** observation-constrained ensemble mean, national trend in the socioeconomic exposure to fire carbon emission in **b** and **c** population, **e** and **f** GDP, and **h** and **i** agricultural area. The trend is presented by the ratio between the projected trend and the exposure during the 2010s (% decade$^{-1}$), indicated by either the **b**, **e**, **h** original or **c**, **f**, **i** observation-constrained ensemble.

emissions and their socioeconomic risks. This observational constraint leads to reduced fire carbon emission magnitudes and a weakened increase in future global fire carbon emission when compared with the original, unconstrained multimodel ensemble (Fig. 3 and Supplementary Fig. 5). The smaller future changes (e.g., both the trends and mean magnitudes) in global fire carbon emission estimated by the observation-constrained multimodel ensemble indicates an overestimated intensity of future wildfires simulated by the default ESMs, and such biases may introduce an overestimation of positive feedbacks of fire to global warming through fire-induced changes in vegetation and soil carbon[42], surface albedo[43], and atmospheric concentrations of aerosols and greenhouse gases[44]. Although the current approach does not account for climate or ecological feedbacks of global fire carbon emissions, dynamical coupling between observation-constrained fire carbon emissions and other components of the Earth system will likely result in a more reliable projection of all these components. The overestimated historical and future enhancement of fire carbon emissions simulated by the default ESMs is mainly distributed across the historically sparsely vegetated regions (Supplementary Fig. 4), potentially as a result of unrealistic representation of dynamic vegetation processes[45]. Furthermore, these ESMs display consistently strong linkage between the simulated historical and future fire carbon emissions (Supplementary Fig. 9), suggesting the primary need to improve historical wildfire simulation for better prediction of future wildfire evolution.

In particular, the observation-constrained ensemble projects increased wildfire activity over the Amazonian and Congo Basins, in contrast to the default simulation for Congo and to a larger extent for Amazon (Fig. 3b, c). The observation-constrained projection of pan-tropical enhancement in wildfire activities is

likely affected by the changes in soil moisture and relative humidity (Supplementary Fig. 10), consistent with previous conclusions regarding accelerated drying over the tropics under climate change[46] and increased occurrence of severe tropical droughts[47–49]. Such apparent association between future drying and elevated fire carbon emission is also identified over other forest, grassland, and cropland, as estimated by the observation-constraint (Supplementary Fig. 11). In the Congo basin, the projected elevation in the amount of fuel[50], partially reflected by the positive contribution of leaf area index trends to the future increased fire carbon emissions as indicated by the observational constraints (Supplementary Fig. 10b), further supports a more flammable future. The leading role of fuel abundance in future fire regimes also appears in other forest, shrubland, savannahs, and cropland (Supplementary Fig. 11). Because of the global, spatial sampling approach (see Methods), our constraining approach results in a much weaker contribution of projected local socioeconomic development (e.g., population density and land use) to the projected trend in fire carbon emissions than the default ensemble, for all major land-cover types (Supplementary Fig. 11). Although the parameterized anthropogenic source and suppression of wildfires in ESMs reflects valuable efforts to represent socioeconomic influence on wildfire regimes, their accuracy and applicability to future scenarios remain to be rigorously evaluated. In this perspective, our MLT-based observation-constrained ensemble raises an alternative scenario of future evolution of fires in the Congo region—with relative weak anthropogenic suppression and/or more anthropogenic ignitions than that estimated by CMIP6 ESMs.

The MLT-based observation-constrained multimodel ensemble projects a further enhancement of wildfire carbon emissions in the historically fire-prone tropical forest ecoregions in western

and equatorial Africa (Fig. 3) and consequently elevated socio-economic risks in these regions (Fig. 4). Owing to their rapid expansion in socioeconomics, including population, GDP, and agriculture, and consequently projected increased vulnerability to wildfires, western and central African countries need to develop mitigation and/or adaptation strategies[3] to minimize potential socioeconomic loss caused by wildfires. Effective wildfire hazard prevention practices may include fuel management through rationally planned prescribed burning[51] and manual modification of vegetation patterns[52], real-time monitoring of smoke spread[53], and accurate forecast of air quality degradation as a result of biomass smoke[54]. For the populated western and northeastern coasts and the Appalachian Mountains of the United States, as well as northern and eastern Australia, our observational constraint confirms the previously projected more flammable future from fuel drying under climate change[55,56], suggesting an increased likelihood of the 2019–2020 Australian bushfire and 2020 extreme western United States wildfire seasons in the upcoming decades. Over these fire-prone regions, projected elevation in mean fire states is likely accompanied with increased occurrence and intensity of extreme wildfire events that may grow beyond suppression capability, thereby requiring a paradigm shift in measuring the effectiveness of fire management policies[57].

The projected fire regimes and their socioeconomic risks depend on the projected socioeconomic pathway. The currently examined SSP5-85 reflects a high-emission scenario[58], whereas a lower-emission scenario, SSP2-45, suggests a generally milder increase in global fire carbon emission, for both the original and observation-constrained ensembles (Supplementary Fig 12). In the northern subtropical and mid-to-high latitudes, while the default ensemble indicates a spatially homogeneous but slightly weaker increase in fire carbon emission in SSP2-45, compared with that estimated for SSP5-85 from the same set of ESMs (Supplementary Fig. 13), the observation-constrained ensemble indicates opposite sign of changes in the Appalachian Mountains of the United States in SSP2-45 and SSP5-85. Greater differences in the projected fire carbon emission between SSP5-85 and SSP2-45, in terms of both sign and magnitude of changes, are seen over the northern subtropics, tropics, and Southern Hemisphere. Such complicated dependency of future projection of fire regimes on socioeconomic pathways is likely attributed to the nonlinear interaction among fire, climate, vegetation, and human activity, as well as potential occurrence of tipping points in ecology and/or climate evolution[59,60].

Several uncertainties and limitations of the current study, as shared by most EC applications[41], are noted here. First, the uncertainty in observational data may evolve into the current observational constraint. Although we analyze a spectrum of data sources for most climatic and ecosystem variables, the single data set used for lightning and socioeconomic variables, as well as the deteriorated reliability of reanalysis-based wind and specific humidity over observation-sparse regions[61], likely leads to a weakened constraint gained from these variables. In addition, the currently examined fire carbon emission data sets were derived from relatively coarse-resolution satellite measurements of burned area (~500 m resolution), which may miss nearly half of the burned area and associated carbon emissions in Africa as detected by higher-resolution satellite measurements (~20 m resolution) in a given year[62]. Inclusion of such small fires in the observational data sets may result in an even greater magnitude of both historical and future global fire carbon emissions estimated by our observational constraint, likely as well as the default ensemble if tuned to match such observation. Second, the inconsistency between observed quantities and model-simulated or model-utilized variables limits further strengthening of our observational constraint. For example, the above ground biomass,

as provided by most ESMs, more directly captures the amount of fuel than the combination of LAI, temperature, and precipitation, as used in our current analytical framework. Yet, a lack of long-term, reliable observational record of above ground biomass prohibits the direct use of such key driving variable in the current analysis. Another example involves the effects of sub-grid topography, such as slope, aspect, and terrain ruggedness at a typical scale of several kilometers, on wildfire spread and intensity[63]. Future ESM development are encouraged to incorporate the sub-grid topographic factors to improve their representation of wildfire regimes and facilitate better observational constraint. Third, the incomplete independence of the analyzed ESMs (e.g., CLM as the terrestrial component of several models) reduces the multimodel spread of the original ensemble, causing higher dependency of our MLT-based EC results on these shared modeling components as well. Moreover, as much as our observation-constrained ensemble provides bias-corrected historical and likely future fire carbon emissions, our approach does not directly account for more complex feedback from socio-economic development to wildfire regimes beyond the prescribed parameterization of socioeconomic drivers of fire carbon emission involved in the currently examined ESMs. Future model development on socioeconomic-wildfire interactions, such as anthropogenic ignition, urbanization, prescribed burning, and anthropogenic suppression on naturel ignitions, will enhance our confidence in predicting future socioeconomic risks from wildfires[11,64,65]. Finally, the current MLT-based observation-constraint framework does not directly account for potential tipping points in fire regime evolution[59,66] or certain threshold in fuel moisture content below which more intense fire behavior may occur[22]. The applicability of our framework to these extreme fire regimes needs further investigation.

In summary, we have developed and applied an MLT-based analytical framework to establish an observation-constrained projection of global fire carbon emissions and socioeconomic exposure using 13 CMIP6 ESMs and multisource, fire-relevant observations. This approach leads to improved representation of both the magnitude and spatial distribution of global fire carbon emission during the validation period. The observation-constrained ensemble projects a 4.1% (2.6%–7.2%) decade$^{-1}$ increase in the global fire carbon emission during the twenty-first century, to a lesser extent than the 6.0% (0.6%–9.4%) decade$^{-1}$ increase as indicated by the default ensemble. Moreover, the observation-constrained ensemble indicates a further enhancement of wildfire carbon emission in the historically fire-prone subtropical savannahs and tropical forests and savannahs in West Africa, Congo, northern Australia, and eastern South America, opposite or to a larger extent than the default ensemble. The rapid development in socioeconomics, including population, GDP, and agriculture in these projected fire-increasing regions results in increased global wildfire risks to population, GDP, and agricultural area by 5.5% (5.0%–6.2%) decade$^{-1}$, 40.6% (33.7%–48.5%) decade$^{-1}$, and 2.5% (1.9%–3.7%) decade$^{-1}$, respectively, during the twenty-first century, which is 39–238% higher than the default ensemble. Such elevated socioeconomic risks from global wildfires are primarily attributed to the concurrently enhanced wildfire activity and socioeconomic exposure across the currently fire-prone West and central Africa during the upcoming decades, calling for mitigation and/or adaptation strategies to minimize potential socioeconomic loss caused by wildfires in these rapidly developing countries. Our MLT-based observational constraining framework provides an encouraging approach for correcting model biases and can be expanded to estimate reasonable evolution of other global and regional climate or ecosystem properties, especially those with extensive local impacts.

## Methods

**Applying traditional EC for global fire carbon emissions.** The recently developed emergent constraint (EC) approach has demonstrated robust capability in reducing the uncertainty in characterizing or projecting Earth system variables simulated by a multimodel ensemble[25,26]. The basic concept of EC is that, despite the distinct model structures and parameters, there exists various across-model relationships (emergent constraints) between pairs of quantities when we analyze outputs from multiple models[27]. Therefore, the EC concept is especially useful to derive the relationship between a variable that is difficult or impossible to measure (e.g., future wildfires) and a second, measurable variable (e.g., historical wildfires), across multiple ESMs. We start with global total values and find significant linear relationship between historical and future global total fire carbon emission across 38 ensemble members of 13 ESMs (Supplementary Fig. 2a). Because we are particularly interested in the spatial distribution of future wildfires, which are critical for quantifying future socioeconomic risks from wildfires, we further apply the EC concept to every grid cell of the globe, using either a single constraint variable (historical fire carbon emissions) or multiple constraint variables (the atmospheric and terrestrial variables in Supplementary Table 2), with the latter being shown in Supplementary Fig. 2b. We find insignificant linear relationships between these historical fire-relevant variables and future wildfires in the historically fire-prone regions across the analyzed 38 members of 13 ESMs. The failure of the traditional EC concept in constraining fire carbon emissions at local scales could be attributed to the highly nonlinear interactions between fire and its cross-section drivers, which is likely inadequately captured by the linear relationship under the EC assumption. Therefore, we further develop an MLT-based constraint to deal with the complex response of wildfires to environmental and socioeconomic drivers.

**MLT-based observational constraining of global fire carbon emissions.** MLT provide powerful tools for capturing the nonlinear and interactive roles among regulators of an Earth system feature, thereby facilitating effective, multivariate constraint on wildfire activity, which represents an integrated function of climate, terrestrial ecosystem, and socioeconomic conditions. MLT have been widely applied for identifying empirical regulators[32] and building prediction systems for global and regional fire activity[35]. To constrain the projected fire carbon emissions simulated by 13 ESMs using observational data, the current study establishes an MLT-based emergent relationship between the future fire carbon emissions and historical fire carbon emissions, climate, terrestrial ecosystem, and socioeconomic drivers.

Here, we use MLT to examine the empirical relationships between historical, observed influencing factors of wildfires and future fire carbon emissions from ESMs and then feed observational data into the trained machine learning models (Supplementary Fig. 3). To train the MLT to use historical states for the prediction of future fire carbon emission, the historical and future simulations from the SSP (Shared Socioeconomic Pathway) 5-85[36], a high-emission scenario, are analyzed for the currently available 13 ESMs in CMIP6 (Supplementary Table 1). A subset of these ESMs (i.e., nine ESMs that provide simulation in a lower-emission scenario, SSP2-45) is also analyzed to examine the dependence of fire regimes on socioeconomic pathway. The training is conducted using the spatial sample of decadal mean predictors and target variable, both individually from each ESM and from their aggregation, with the later referred to as multimodel mean and subsequently analyzed for projecting fire carbon emission and its socioeconomic risks. Corresponding to the spatial resolution of the observational products of fire carbon emission, all model outputs are bilinearly interpolated to a $0.25° \times 0.25°$ grid, resulting in a spatial sample of 11,325 points per model for the training. To perform the observational constraint, the historical observed predictors are then fed into the trained machine learning models. The historical predictors are listed in Supplementary Table 2 with their observational data sources, temporal coverages, and spatial resolutions. For the atmospheric and terrestrial variables, the annual mean value and climatology in each of 12 calendar months are included as predictors. This training and observational constraining is performed for target decades (2011–2020, 2021–2030,… 2091–2100), and the historical period is always 2001–2010. Future changes in fire carbon emission are quantified and expressed as the relative trend (% decade$^{-1}$) (i.e., the ratio between the absolute trend and the mean value during the 2010s), for both the default and observation-constrained ensembles.

The current spatial sample training approach establishes a history-future relationship for each pixel using the entire global sample. To minimize local prediction errors for a certain pixel, MLT search all pixels, regardless of their geographical location, to optimize the prediction model of future fires at the target pixel. In this way, a physically robust history-future relationship is established based on the global sample of locations, whereas influences of localized features, such as socioeconomic development, on wildfire trends are naturally damped in our approach (Supplementary Figs. 10 and 11). The reliability of MLT is degraded when the actual observational data space is insufficiently covered by the training (historical CMIP6 simulation) data space, namely the extrapolation uncertainty. Here, we further evaluate the data space of both observation and historical simulation of the climate and fire variables (Supplementary Fig. 14), and we find all these assessed variables are largely overlapped, indicating minimal extrapolation error involved in the current MLT application.

To minimize the projection uncertainty associated with the selected machine learning algorithms, this study examines three MLT—random forest (rf), support vector machine with Radial Basis Function Kernel (svmRadialCost), and gradient boosting machine (gbm). These three algorithms differ substantially in their function. The average among these algorithms is thus believed to better capture the complex interrelation between the historical predictors and future fire carbon emissions than any single algorithm. The MLT analysis is performed using the "caret," "dplyr," "randomForest," "kernlab," and "gbm" packages in the R statistical software. The prediction model is fitted for each MLT using the training data set that targets each future decade, with parameters optimized for the minimum RMSE via 10-fold cross-validation—in other words, using a randomly chosen nine-tenth of the entire spatial sample ($n = 10,193$) for model fitting and the remaining one-tenth of the entire spatial sample ($n = 1,132$) for validation, and repeating the process 10 times. For svmRadialCost, the optimal pair of cost parameter (C) and kernel parameter sigma (sigma) is searched from 30 (tuneLength = 30) C candidates and their individually associated optimal sigma. For gbm, we set the complexity of trees (interaction.depth) to 3, and learning rate (shrinkage) to 0.2, and let the "train" function search for the optimal number of trees from 10 to 200 with an increment of 5 (10, 15, 20, …, 200). For rf, the number of variables available for splitting at each tree node (mtry) is allowed to search between 5 and 50 with an increment of 1 (5, 6, 7, …, 50); the number of trees is determined by the algorithm provided by randomForest package and the "train" function by the caret package. The cross-validation $R^2$s exceed 0.8 ($n = 1,132$) for all optimized MLT and all future periods. The currently examined ESMs, MLT, and hundreds of observational data set combinations constitute a multimodel, multi–data set ensemble of projected fire carbon emissions for the twenty-first century. This multimodel, multi–data set ensemble allows natural quantification of uncertainty in the future projection derived from observational sources and MLT, compared with a previous single-MLT, single-observation approach[67].

This MLT-based observational constraining approach is validated for a historical period using the emergent relation between the fire-climate-ecosystem-socioeconomics during 1997–2006 and fire carbon emission during 2007–2016. The spatial correlation and RMSE with the observed decadal mean fire carbon emission ($n = 11,325$) is evaluated and compared for the constrained and unconstrained ensemble, reported in the main text (Figs. 1 and 2). The RMSE and $R^2$ produced by the traditional EC approach that constrains fire carbon emissions during 2007–2016 with fire carbon emissions during 1997–2006 are reported along with the MLT-based observational constraint in Fig. 1e, f. The MLT-based observational constraining approach is also applied to six ESMs that report burned area fraction, and validation is also conducted and reported in Supplementary Fig. 6.

Because the MLT are trained using the global spatial sample, we expect the performance of MLT to be sensitive to the spatial resolution of the training data set. This assumption is tested by varying the interpolation grids (1°, 2.5°, 5°, and 10° latitude by longitude) of the ESMs and fitting MLT using this specific-resolution training data for the validation period (Supplementary Fig. 7). Observational data sets at 0.25° resolution are subsequently fed into the fitted MLT models, regardless of the input model data resolution. This sensitive test sheds light on the importance of spatial resolution to our observational constraining and thereby implies potential accuracy improvement of our MLT-based observation constraint with the development of higher-resolution ESMs.

**Socioeconomic risks from fire carbon emission.** Here, we define the socioeconomic exposure to wildfires as a product of decadal mean fire carbon emission and number of people, amount of GDP, and agricultural area exposed to the burning in each grid cell, following previous definition for extreme heat[68]. These exposure metrics measure the amount of population, GDP, and agricultural area affected by wildfires, whose severity is represented by the amount of fire carbon emission. The projected population at $1/8° \times 1/8°$ resolution under SSP5-85 is obtained from the National Center for Atmospheric Research's Integrated Assessment Modeling Group and the City University of New York Institute for Demographic Research[69]. The projected GDP at 1 km resolution under SSP5 is disaggregated from national GDP projections using nighttime light and population[70]. The agricultural area projection at $0.05° \times 0.05°$ resolution under SSP5-85 is obtained from the Global Change Analysis Model and a geospatial downscaling model (Demeter)[71]. All the projected socioeconomic variables are resampled to $0.25° \times 0.25°$ resolution before the calculation of exposure to fire carbon emission fraction. Future changes in socioeconomic exposure to wildfires are quantified as the relative trend (% decade$^{-1}$) (i.e., the ratio between the absolute trend and the mean value during the 2010s) for the default and observation-constrained ensembles. These relative changes provide direct implications on what the future would be like compared with the current state, regardless of the potential biases simulated by the default ESMs.

**Understanding projected wildfire trends through importance scores reported by MLT.** The mechanisms underlying the projected evolution in fire carbon emissions are explored in two tasks, addressing the importance of drivers in the historical and dynamical perspectives. The first task assesses the relative contribution of each environmental and socioeconomic driver's historical distribution to the projected future wildfire distribution, for directly understanding how the current observational constraint works (Supplementary Fig. 8). The second task examines the relative contribution of each driver's projected trend to the projected wildfires trends in a specific region, for disentangling the dynamical mechanisms

underlying future evolution of regional wildfires (Supplementary Fig. 9). These tasks benefit from the importance score as an output of MLT. Although the calculation of importance scores varies substantially by MLT, all the importance scores qualitatively reflect relative importance of each predictor when making a prediction. For each tree in both rf and gbm, the prediction accuracy on the out-of-bag portion of the data is recorded. Then, the same is done after permuting each predictor variable. For rf, the differences are averaged for each tree and normalized by the standard error. For gbm, the importance order is first calculated for each tree and then summed up over each boosting iteration. For svm, we estimate the contribution of a single variable by training the model on all variables except that specific variable. The difference in performance between that model and the one with all variables is then considered the marginal contribution of that particular variable; such marginal contribution of each variable is standardized to derive the variable's relative importance. Because we apply multiple MLT in this study, the average importance scores from these MLT are reported in the corresponding figures for robustness.

In the first task, the importance of each historical driver to future global wildfire distributions is examined in three MLT models (random forest, support vector machine, and gradient boosting machine) that are trained for projecting future fire carbon emissions (Supplementary Fig. 8). For the atmospheric and terrestrial variables that include annual mean and monthly climatology as predictors, to account for the overall importance of a particular variable while considering the possible information overlapping contained in each month and annual mean, the importance of each variable is represented by the highest importance score among these 13 predictors (annual mean, January, February, …, December). The importance score of each historical driver reflects the relative weight of each historical, environmental driver in determining the spatial pattern of fire carbon emissions in each future decade.

In the second task, the dynamical importance of each environmental driver's future evolution is assessed for targeted tropical regions (i.e., Amazon and Congo) and major land cover types (tropical forests, other forest, shrubland, savannas, grasslands, and croplands) in both default and constrained ensembles through the importance of each driver's trend to the projected wildfire trend. For the default ensemble, the three MLT models (random forest, support vector machine, and gradient boosting machine) are used to predict the spatial distribution of simulated trends in fire carbon emission using the simulated trends in the socioeconomic, atmospheric, and terrestrial variables that are considered in our observational constraint for wildfires, for each ESM and their multimodel mean. This analysis excludes flash rate, another predictor in constraining future wildfires, because it is not dynamically simulated by most ESMs. For the observation-constrained ensemble, we first constrain the projected atmospheric and terrestrial variables in each future decade, using a similar approach as we constrain future fire carbon emissions, for each individual ESM and their multimodel aggregation. In this constraint for environmental drivers, all the variables in Supplementary Table 2 are considered as predictors, thereby achieving self-consistency of the constrained future evolution of all these fire-relevant variables. Noticing that the socioeconomic trends are determined by the SSPs, future socioeconomic developments are therefore not constrained in the current approach. Then, the same three MLT models are used to predict the spatial distribution of constrained trends in fire carbon emissions using the constrained trends in those environmental and socioeconomic drivers. For computational efficiency, only the annual mean trends in the environmental drivers are constrained and analyzed in this task. The importance scores of projected trends in socioeconomic and environmental drivers reflect their dynamic role in future evolution of wildfires in the target tropical regions. Here, the Amazon and Congo regions are shown as examples of how this analysis is applied to understand regional wildfire evolutions, though the mechanism underlying the future evolution of wildfires in other regions could be similarly explored.

## Data availability

CMIP6 model outputs can be openly accessed via different Earth System Grid Federation (ESGF) data nodes (e.g., https://esgf-node.llnl.gov/projects/cmip6/). The observational datasets utilized in this study are derived from published sources, cited in the Supplementary Table 2. The data that support the plots within this paper and other findings of this study are available from the corresponding author upon request.

## Code availability

The code to carry out the current analyses is available from the corresponding author upon request.

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

## Acknowledgements

This work is supported by funding provided by the Environmental Sciences Division at the US Department of Energy's (DOE's) Oak Ridge National Laboratory (ORNL), and partially supported by the Reducing Uncertainties in Biogeochemical Interactions through Synthesis and Computing Scientific Focus Area (RUBISCO SFA) project and the Terrestrial Ecosystem Science Scientific Focus Area (TES SFA) project funded through the Earth and Environmental Systems Sciences Division of the Biological and Environmental Research Office in the DOE Office of Science. ORNL is supported by the Office of Science of the DOE under Contract No. DE-AC05-00OR22725. Computation is supported by High-performance Computing Platform of Peking University.

## Author contributions

J.M. conceived the research; Y.Y. and J.M. performed the analysis and drafted the paper; and Y.Y., J.M., S.W., A.C., X.S., Y.W., F.H., Y.Z., and E.P. wrote the paper.

## Competing interests
The authors declare no competing interests.
