## [Peer Review File · Nature Communications]

Peer review comments, first round review–

Reviewer #1 (Remarks to the Author):

The work "Machine learning-based observation-constrained projections reveal elevated global socioeconomic risks to wildfire in the twenty-first century" presents a very promising method to forecast wildfire impacts and its associated risks. The authors propose a new methodological approach combining ESM and Machine Learning to introduce constraints about fire-related drivers in the modeling workflow. The results suggest a clear improvement when compared with "original" methods both in terms of overall magnitude of CO₂ emission and its spatial pattern. The procedure includes not only a validation that sets a common ground for comparison but several means to illustrate the main differences and improvements. Overall, the conclusions are well aligned by the results, though additional clarifications and modifications are required. The methodology couples together ESMs and MLT in an effort to overcome some limitations of the current procedures. The main advantage is that MLT can embed non-linear feedbacks from wildfire drivers. The concept looks promising. The description of the methods is quite complete though further details need to be provided in order for the method to be fully reproduced. The ESM models and input data are well described. Likewise, the MLT and associated packages are also presented together with a summary of the overall workflow for validations and performance evaluation.

In spite of the promising results that will surely contribute to the current state-of-the-art, there are some limitations that I believe require further investigation.

1) In lines 196-197 it is stated that "four ESMs... ...simulate a reduced global fire carbon emission during the first half of the twenty-first century". This seems rather unrealistic and partially responsible for the relatively low increase in emissions towards the end of the century (2.5% according to the proposed method compared to the 6% in the original one). The issue is clearly related to the 2 NorESM models, pulling down fire-carbon emissions during 2040-2050. The predicted drop is equal in magnitude the increase from 2050 to 2100 while setting emissions below 2010's estimates. What would be the emission estimates disregarding these ESMs? Moreover, the sudden decay between 2040 and 2050 seems to be what drives risk trajectories (Figure 3a and 3g), which would otherwise be more similar.

2) Another major concern has to do with the use of MLT. More specifically to the possibility the limited capability of MLT (especially Random Forest) to extrapolate outside the range of observations is hindering the projections.

3) In the same line, how does the model respond to a more conservative scenario? What's the difference in magnitude compared to an SSP2-4.5, for instance? My main concern here has to do with the sensitivity of the proposed method to properly address different scenarios. Please, can you elaborate on this?

4) Three MLT with very different parameterization were used. While random forest is relatively easy to tune up, support vector machines are quite challenging to optimize. Further insights into model calibration are required. Since caret is involved I assume parameter grids were fed to optimize parameter selection. I understand this is not a "manual" on how to calibrate ML models but the very basics must be provided. E.g., number of trees for RF, kernel types in SVM, learning rates. Where these parameters set constant over time?

Moreover, how MLT were combined? I mean, how were predictions from the three models integrated? Is there any kind of "model averaging" involved?

What are these importance metrics used in each MLT? I am not sure the comparison between different metrics is reliable even in relative terms. There are examples of performance assessment through jackknife estimators using a common performance metric from test samples fitting univariate or models without a candidate predictor to assess the sensitivity of the model to that predictor. See for instance:

Bar Massada, A., Syphard, A.D., Stewart, S.I., Radeloff, V.C., 2012. Wildfire ignition-distribution modelling: a comparative study in the Huron–Manistee National Forest, Michigan, USA. *Int. J. Wildl. Fire* 22, 174–183. <https://doi.org/http://dx.doi.org/10.1071/WF11178>

Minor concerns

Abstract. References are not common in an abstract and I don't think the ones included are critical. They are cited right away in the Introduction.

L53. Consider replacing "global fire regimes" with "fire regimes across the globe" or something similar. There is not actual global fire regimes but a collection of fire regimes and pyroregions.

L82. "reducing uncertainties". I would say that also spatial inaccuracies judging by the outputs presented. One of the most striking improvements come from the improved capability of the method to properly capture the spatial patterns. See Fig. 1 for example.

L151-152. About the model performance. Some insights into the bias (over vs underestimation) and it's spatial footprint would enrich the findings.

L155. "more optimal" with "better"

L177-178. Was this calculated at pixel level or are R2 calculated for each ESM? If it is the first, acknowledging the regions with weaker performances will improve the understanding of the outputs.

L284. Africa is not a county. Rephrase please.

L350-352. Actually the original approach shows a clear upward trend in emissions whereas the constrained model fluctuates over time (2040-2050) that can be consider a large relative change. The small changes are driven by the EMS that forecast a drop in that period but the differences disregarding those models wouldn't that small.

L371. Fuel, not biofuel.

L382-383. Actually, most studies envisage that future extreme wildfires might be beyond suppression capability, regardless of the means available. See:

Moreira, F., Ascoli, D., Safford, H., Adams, M.A., Moreno, J.M., Pereira, J.M.C., Catry, F.X., Armesto, J., Bond, W., González, M.E., Curt, T., Koutsias, N., McCaw, L., Price, O., Pausas, J.G., Rigolot, E., Stephens, S., Tavsanoğlu, C., Vallejo, V.R., Wilgen, B.W. Van, Xanthopoulos, G., Fernandes, P.M., 2020. Wildfire management in Mediterranean-type regions: paradigm change needed. *Environ. Res. Lett.* 15, 11001. <https://doi.org/10.1088/1748-9326/ab541e>

L406-408. Can this connect with the more conservative estimates in fire-related emissions in the future??

L413. But this also increase the estimate in the original model, right?

L507-508. How is this similarity gauged?

L545. Rather increased observational data improving the spatial accuracy of the interpolation.

L561. Interpolated or resampled?

L588. I think average or median values plus standard deviation or IQR are more appropriate.

Reviewer #2 (Remarks to the Author):

Yu et al NCC emergent constraints method for wildfire

The authors perform a variation of an emergent constraints analysis on earth system model projections of wildfire emissions under climate change. They use multiple fire relevant variables in combination with several machine learning techniques to constrain current and future estimates of the emissions from wildfire around the globe. They also estimate the exposure of population, GDP and agriculture to changes in emissions. Their technique does indeed adjust model outputs closer to observations, and results in a narrower, and generally lower estimate of the trajectory of future wildfire emissions, albeit one that is still rising steadily throughout the 21st century.

Congratulations to the authors on an interesting and potentially useful study on an important topic. Improving projections of global wildfire activity is a very worthy task and the paper is generally well organised with good figures. Applying the emergent constraints paradigm to wildfire modelling is novel, as is the use of multiple variables and a machine learning approach. I have some high level concerns, as well as some more specific ones, which I detail below.

- In my opinion it is critical that an expert on EC be included among the reviewers. I am not an expert on emergent constraints, but it is not clear to me that the authors have avoided the pitfalls associated with this approach. Some of these are outlined in Hall et al. 2019, Williamson et al. 2021, Sanderson et al. 2021.

o Strength of the statistical relationship. As far as I can tell Supp Fig 2 is the only place that the actual performance of the model is displayed. While it is a significant improvement on the 'default' EC option there are still large areas of fire-prone land with no significant correlation. This includes quite large parts of Australia, South America, Africa, North America and Europe, many of which are discussed at length in the manuscript. Can the authors clarify the amount of flammable land/burned area/emissions that their model is reliable for? Is it valid to apply the model in areas with no significant relationship? And does it matter that the model reproduces mean global emissions accurately in spite of these large areas without correlation? Could errors be cancelling each other out? Given the importance of this step, it is not clear to me that the authors have established the relationship significantly. They could perhaps provide some information to evaluate the machine learning approaches they have taken, either individually or in aggregate. No doubt an expert on EC could shed light on many of these issues.

o Insufficient sample size – is the use of individual pixels enough to overcome this?

o Model independence – multiple models share the same land use and fire parametrisations

o Overlooking the potential for tipping points – these are widely recognised to be a risk for some fire regimes (Lenton et al. 2008)

o The mechanisms underpinning the relationship. I applaud the authors for including multiple relevant variables, but the use of machine learning largely obscures any mechanisms at play (the variable importance plots in the Supp Info notwithstanding). The authors describe this as process-based but I'm not sure this qualifies, as there is no explicit process. The authors do not provide any information on how they selected those variables (L518-520), or whether a different set might have performed better. Given the centrality of fire to this study, it would be good to see this addressed.

o Quality of observations. It is worth acknowledging limitations in this data. Wind speed in particular suffers from a lack of long term high quality observational datasets against which to judge the accuracy of reanalysis values. Relative humidity, flash rate and soil moisture are probably also a fair way behind variables like temperature and rainfall in terms of their accurate simulation. See L409

o Can you clarify that you have avoided the possibility of strong but "overconfident" constraints (Sanderson et al. 2021)

o Can you give an indication of reasonable performance of EC and its basis? Presumably the process by definition improves accuracy, so what is the best way to judge its effectiveness?

- There are a number of simple errors in language and referencing which can easily be addressed, but somewhat undermine confidence in the rest of the manuscript e.g.

o It is risks from / posed by wildfire, not risks to wildfire

o Some references are listed twice

o L91 bracket in wrong spot

o Abstract: Forkel et al. and Li et al. are good references but do not go to the claim that it is a lack of observational constraints that limit wildfire projection credibility. It is arguably a lack of process-based understanding that limits confidence on ESM wildfire simulation (Boer et al. 2021)

o Language is a little too strong e.g. drastically, extremely disastrous, enormous, huge, remarkably, vast etc

o Use of Wikipedia as a reference (I am a big fan of Wikipedia but don't think it is appropriate as a source)

o The reference to Abatzoglou et al (#7) is about fire weather, not fire behaviour

o Similarly, references #11 and #12 are about projections, whereas your claim is that changes have already occurred

o This may be a language issue, but refs #3 and #4 do not talk about feedbacks. Likewise #13 does not say that weaknesses in feedback representation lead to biases, it is more of a snap evaluation than a thorough one.

o #8 does not seem to actually discuss "complex, nonlinear integration of meteorological, ecological, and socioeconomic states"

o #43 would be better off being a general paper on prescribed burning rather than a projection of

prescribed burning conditions e.g. Russell-Smith et al 2020.

o #45 is about smoke not fire danger

o L320 should this be Supp Fig 2?

o L343 you cannot say that you have made an accurate assessment of future risk. That remains to be seen

o L367 onwards. I would be a bit more cautious in my language here, as you are speculating about the drivers of your results rather than actually testing them.

o #48 lists the wrong authors

Other comments

- L73-78 can you rephrase to make this clearer? I'm not sure what you mean by sections or multisector states, nor how the listed references relate to this statement.

- L126 A good citation is Jain et al. 2020

- If possible it would be good to show a map of actual emissions (rather than % change) in the Supp Info (equivalent to Fig 2b,c,d)

- Likewise can the authors please show model bias e.g. Fig 1, Supp Fig 6 – don't just show model and observations, but show the difference between the two

- It is important to acknowledge that SSP-5 is not business as usual, but a (fortunately) less likely higher emissions scenario (Hausfather & Peters 2020). Ideally the authors would include a more realistic scenario as well, but at the least they could acknowledge this.

- It's great that the authors include information about the fire model within ESMs in Supp Table 1, but these descriptions don't mean much to me or I suspect most readers. Can the authors expand this to a short para, or in some other way clarify the general nature of these fire modules and how they differ?

- The country based approach is definitely valuable (Fig 3) as is the Congo and Amazon zoom in (Supp Fig 9). However, I think you will find better relationships and more meaningful drivers if you focus on dominant fire types e.g. forest fire, grass fire, peats etc.

References

- Boer et al 2021 A Hydroclimatic Model for the Distribution of Fire on Earth
<https://doi.org/10.1088/2515-7620/abec1f>

- Hall et al. 2019 Progressing emergent constraints on future climate change
<https://doi.org/10.1038/s41558-019-0436-6>

- Hausfather & Peters 2020 Emissions – the 'business as usual' story is misleading.
<https://doi.org/10.1038/d41586-020-00177-3>

- Jain et al. 2020 A review of machine learning applications in wildfire science and management.
<http://dx.doi.org/10.1139/er-2020-0019>

- Lenton TM, Held H, Kriegler E, Hall JW, Lucht W, Rahmstorf S and Schellnhuber HJ 2008 Tipping elements in the Earth's climate system Proc. Natl. Acad. Sci. USA 105 1786–93

- Russell-Smith et al 2020 Adaptive prescribed burning in Australia for the early 21st Century – context, status, challenges <https://doi.org/10.1071/WF20027>

- Sanderson et al. 2021 The potential for structural errors in emergent constraints
<https://doi.org/10.5194/esd-12-899-2021>

- Williamson et al. 2021 Emergent constraints on climate sensitivities
<https://doi.org/10.1103/RevModPhys.93.025004>

Reviewer #3 (Remarks to the Author):

[See next page]

Review of NCOMMS-21-34508-T “Machine learning-based observation-constrained projections reveal elevated global socioeconomic risks to wildfire in the twenty-first century” by Yu et al.

This manuscript considers the use of machine learning techniques (MLTs) to better represent the nonlinear relationships between fire occurrence and various environmental factors at global scales, and to then use this to predict carbon release and socioeconomic impacts. Fire has traditionally been poorly represented in Earth system/global climate models and so studies like this one, which aim to improve the representation of fire are vitally important in providing an accurate picture of global change. The use of MLTs to better express the nonlinear relationships between interacting aspects of complex systems had success in other contexts, and so while their use is not new, the application to global fire occurrence and carbon release is particularly innovative. As such, the manuscript addresses a significant issue, and I am sure it will attract considerable scientific interest.

While my overall opinion is that the manuscript is worthy of publication, I feel that the manuscript could be strengthened by addressing a few issues that are not adequately addressed in the current version. These are as follows:

1. The authors mention the 2019-20 Australian fires, which are notable for the massive amount of carbon they released. Much of this occurred in connection with numerous episodes of extremely intense fire behaviour. These episodes have been linked to particular environmental factors, primarily rugged terrain, forest fuels and critically low fuel moisture content, and their interaction (e.g., Sharples et al. 2016; Di Virgilio et al. 2019; Abram et al. 2021). While it could be argued that land-use accounts for forest fuels and that fuel moisture content is somewhat covered by temperature and relative humidity, the list of variables considered by the authors (Extended Data Table 2) doesn't include any terrain variables. I'm therefore left to wonder whether the author's analyses have identified all the relationships that are important to carbon release – particularly massive carbon release events.
2. How does Leaf Area Index (LAI) account for the extensive regions of grasslands, savannah and other important vegetation types around the globe (e.g., xeric shrublands, spinifex, etc.)? Moreover, how does satellite-derived LAI account for fuels in the surface and near-surface layers of forests, noting that it's the fuels within these layers (and their dryness) that have the greatest influence on fire occurrence and account for a considerable proportion of carbon release?
3. Fuel moisture, which depends on air temperature and relative humidity, can have a significant effect on wildfire behaviour. For example, Abram et al. (2021)

note that when fuel moisture content drops below certain thresholds, wildfire behaviour can be noticeably different. This is due to greater propensity for certain types of fire behaviour, such as spotting. My concern in this respect, is that these thresholds are not incorporated in the approach used by the authors, and so their analyses may be missing important dynamics that could occur more often in the future as conditions become warmer. In particular, the methods used may not account for more intense fire behaviour, which could release substantially more carbon. The authors should note this as a limitation of their study and revise their conclusions accordingly.

4. The release of carbon from global fire constitutes an important climate feedback. Was this accounted for in the study? That is, were the carbon release estimates found in the study used to further inform the climatic projections used in a coupled manner? Again, if not, this should be noted as a limitation of the study.

I think that once these issues have been addressed the manuscript will be suitable for publication. However, I think doing so might constitute a major revision of the manuscript.

References

Abram, N.J., Henley, B.J., Gupta, A.S., Lippmann, T.J., Clarke, H., Dowdy, A.J., Sharples, J.J., Nolan, R.H., Zhang, T., Wooster, M.J. and Wurtzel, J.B., 2021. Connections of climate change and variability to large and extreme forest fires in southeast Australia. *Communications Earth & Environment*, 2(1), pp.1-17.

Di Virgilio, G., Evans, J.P., Blake, S.A., Armstrong, M., Dowdy, A.J., Sharples, J. and McRae, R., 2019. Climate change increases the potential for extreme wildfires. *Geophysical Research Letters*, 46(14), pp.8517-8526.

Sharples, J.J., Cary, G.J., Fox-Hughes, P., Mooney, S., Evans, J.P., Fletcher, M.S., Fromm, M., Grierson, P.F., McRae, R. and Baker, P., 2016. Natural hazards in Australia: extreme bushfire. *Climatic Change*, 139(1), pp.85-99.

Reviewers' comments:

Reviewer #1 (Remarks to the Author):

The work “Machine learning-based observation-constrained projections reveal elevated global socioeconomic risks to wildfire in the twenty-first century” presents a very promising method to forecast wildfire impacts and its associated risks. The authors propose a new methodological approach combining ESM and Machine Learning to introduce constraints about fire-related drivers in the modeling workflow. The results suggest a clear improvement when compared with “original” methods both in terms of overall magnitude of CO₂ emission and its spatial pattern. The procedure includes not only a validation that sets a common ground for comparison but several means to illustrate the main differences and improvements. Overall, the conclusions are well aligned by the results, though additional clarifications and modifications are required. The methodology couples together ESMs and MLT in an effort to overcome some limitations of the current procedures. The main advantage is that MLT can embed non-linear feedbacks from wildfire drivers. The concept looks promising. The description of the methods is quite complete though further details need to be provided in order for the method to be fully reproduced. The ESM models and input data are well described. Likewise, the MLT and associated packages are also presented together with a summary of the overall workflow for validations and performance evaluation. In spite of the promising results that will surely contribute to the current state-of-the-art, there are some limitations that I believe require further investigation.

Response: Thank you very much for your encouragement and suggestions. We hope our additional analysis strengthens this manuscript.

1) In lines 196-197 it is stated that “four ESMs... simulate a reduced global fire carbon emission during the first half of the twenty-first century”. This seems rather unrealistic and partially responsible for the relatively low increase in emissions towards the end of the century (2.5% according to the proposed method compared to the 6% in the original one). The issue is clearly related to the 2 NorESM models, pulling down fire-carbon emissions during 2040-2050. The predicted drop is equal in magnitude the increase from 2050 to 2100 while setting emissions below 2010’s estimates. What would be the emission estimates disregarding these ESMs? Moreover, the sudden decay between 2040 and 2050 seems to be what drives risk trajectories (Figure 3a and 3g), which would otherwise be more similar.

Response: Thank you for raising this issue. In the revised manuscript, we include an additional predictor, orography, as suggested by another reviewer. We also include an additional observational data set of soil moisture. These changes lead to the updated results on global fire carbon emission prediction (Fig. 2) and associated socioeconomic risks (Fig. 3). As you can see, in the updated projection, the influence of the two NorESM models partially diminishes. To quantify the sensitivity of observation-constrained projection to the NorESM models, we repeat the analysis with a subset of the CMIP6 ESMs that exclude the two NorESM models (Extended Data Fig. 4). Relevant results are described in the revised manuscript: “*However, four ESMs (CESM2, CESM2-WACCM, NorESM2-LM, and NorESM2-MM) that share the same land component, namely the Community Land Model (CLM) version 5 (Extended Data Table 1), simulate a reduced global fire carbon emission during the first half of the twenty-first century, resulting in a relatively stable fire carbon emission from 2010s to 2050s produced by the observation-constrained ensemble (Fig. 2a). An exclusion of the NorESM2 models leads to a slightly elevated future increase in fire carbon emissions produced by the observational constraint, especially over the northern extratropical land surface (Extended Data Fig. 4).*” (page 8 lines 201–208)

2) Another major concern has to do with the use of MLT. More specifically to the possibility the limited capability of MLT (especially Random Forest) to extrapolate outside the range of observations is hindering the projections.

Response: We agree that extrapolation remains one of the key uncertainties imbedded in the implementation of MLT. Our analytical framework uses CMIP6 historical results as input and CMIP6 future simulations as output to train three selected MLT, and then feeds observational data into the trained MLT (Methods and Extended Data Fig. 3). Therefore, potential extrapolation occurs when the observational data space of predictors is not fully covered by the multimodel simulation of historical data space for these predictors. To demonstrate the data space coverage of observations and multimodel simulations, we add scatterplots of observational and simulated predictors in Extended Data Fig. 14. The observational data space is mostly covered by the multimodel historical simulation, thereby ensuring minimal extrapolation uncertainty involved in the MLT. We include a brief description on the uncertainty source in the revised manuscript: *“The reliability of MLT is degraded when the actual observational data space is insufficiently covered by the training (historical CMIP6 simulation) data space, namely the extrapolation uncertainty. Here, we further evaluate the data space of both observation and historical simulation of the climate and fire variables (Extended Data Fig. 14), and we find all these assessed variables are largely overlapped, indicating minimal extrapolation error involved in the current MLT application.”* (pages 20–21 lines 557–562)

3) In the same line, how does the model respond to a more conservative scenario? What's the difference in magnitude compared to an SSP2-4.5, for instance? My main concern here has to do with the sensitivity of the proposed method to properly address different scenarios. Please, can you elaborate on this?

Response: Thank you for this helpful comment. To demonstrate the applicability of the current approach addressing different scenarios, we apply our framework onto nine CMIP6 ESMs that provide SSP2-45 simulations (Extended Data Fig. 12) and compare the results with the analysis conducted for SSP5-85 with the same set of models (Extended Data Fig. 13). The results and implications are discussed in the revised manuscript: *“The projected fire regimes and their socioeconomic risks depend on the projected socioeconomic pathway. The currently examined SSP5-85 reflects a high-emission scenario⁵⁸, whereas a lower-emission scenario, SSP2-45, suggests a generally milder increase in global fire carbon emission, for both the original and observation-constrained ensembles (Extended Data Fig 12). In the northern subtropical and mid-to-high latitudes, while the default ensemble indicates a spatially homogeneous but slightly weaker increase in fire carbon emission in SSP2-45, compared with that estimated for SSP5-85 from the same set of ESMs (Extended Data Fig. 13), the observation-constrained ensemble indicates opposite sign of changes in the Appalachian Mountains of the United States in SSP2-45 and SSP5-85. Greater differences in the projected fire carbon emission between SSP5-85 and SSP2-45, in terms of both sign and magnitude of changes, are seen over the northern subtropics, tropics, and Southern Hemisphere. Such complicated dependency of future projection of fire regimes on socioeconomic pathways is likely attributed to the nonlinear interaction among fire, climate, vegetation, and human activity, as well as potential occurrence of tipping points in ecology and/or climate evolution^{59,60}.”* (page 16 lines 421–434)

4) Three MLT with very different parameterization were used. While random forest y relatively easy to tune up, support vector machines are quite challenging to optimize. Further insights into model calibration are required. Since caret is involved I assume parameter grids were fed to optimize parameter selection. I understand this is not a “manual” on how to calibrate ML models but the very basics must be provided. E.g., number of trees for RF, kernel types in SVM, learning rates. Where these parameters set constant over time?

Response: Thank you for this helpful suggestion. The parameter optimization is now briefly discussed in the Methods section: *“The prediction model is fitted for each MLT using the training data set that targets each future decade, with parameters optimized for the minimum RMSE via 10-fold cross-validation—in other words, using a randomly chosen nine-tenth of the entire spatial sample ($n = 10,193$) for model fitting and the remaining one-tenth of the entire spatial sample ($n = 1,132$) for validation, and repeating*

the process 10 times. For svmRadialCost, the optimal pair of cost parameter (C) and kernel parameter sigma (sigma) is searched from 30 (tuneLength = 30) C candidates and their individually associated optimal sigma. For gbm, we set the complexity of trees (interaction.depth) to 3, and learning rate (shrinkage) to 0.2, and let the “train” function search for the optimal number of trees from 10 to 200 with an increment of 5 (10, 15, 20, ..., 200). For rf, the number of variables available for splitting at each tree node (mtry) is allowed to search between 5 and 50 with an increment of 1 (5, 6, 7, ..., 50); the number of trees is determined by the algorithm provided by randomForest package and the “train” function by the caret package.” (page 21 lines 570–582)

Moreover, how MLT were combined? I mean, how were predictions from the three models integrated? Is there any kind of "model averaging" involved?

Response: Yes, we simply average the results from different MLT in this study, as clarified in the revised manuscript. (page 23 lines 649–650)

What are these importance metrics used in each MLT? I am not sure the comparison between different metrics is reliable even in relative terms. There are examples of performance assessment through jackknife estimators using a common performance metric from test samples fitting univariate or models without a candidate predictor to assess the sensitivity of the model to that predictor. See for instance:

Bar Massada, A., Syphard, A.D., Stewart, S.I., Radeloff, V.C., 2012. Wildfire ignition-distribution modelling: a comparative study in the Huron–Manistee National Forest, Michigan, USA. *Int. J. Wildl. Fire* 22, 174–183. <https://doi.org/http://dx.doi.org/10.1071/WF11178>

Response: Thank you for raising this question. In the revised manuscript, we briefly introduce the importance metric for each algorithm as detailed here: *“For each tree in both rf and gbm, the prediction accuracy on the out-of-bag portion of the data is recorded. Then, the same is done after permuting each predictor variable. For rf, the differences are averaged for each tree and normalized by the standard error. For gbm, the importance order is first calculated for each tree and then summed up over each boosting iteration. For svm, we estimate the contribution of a single variable by training the model on all variables except that specific variable. The difference in performance between that model and the one with all variables is then considered the marginal contribution of that particular variable; such marginal contribution of each variable is standardized to derive the variable’s relative importance.”* (page 23 lines 641–649)

Minor concerns

Abstract. References are not common in an abstract and I don’t think the ones included are critical. They are cited right away in the Introduction.

Response: Thank you; references were removed from the Abstract.

L53. Consider replacing “global fire regimes” with “fire regimes across the globe” or something similar. There is not actual global fire regimes but a collection of fire regimes and pyroregions.

Response: Thank you; we have changed “global fire regimes” to “fire regimes across the globe” throughout the manuscript.

L82. “reducing uncertainties”. I would say that also spatial inaccuracies judging by the outputs presented. One of the most striking improvements come from the improved capability of the method to properly capture the spatial patterns. See Fig. 1 for example.

Response: Thank you; the relevant sentence has been changed to “...we hypothesize that constraining the ESM wildfire estimates by observations is a potentially valid approach for reducing spatial inaccuracies in global wildfire projections and related socioeconomic risks.” (page 3 lines 74–76)

L151-152. About the model performance. Some insights into the bias (over vs underestimation) and its spatial footprint would enrich the findings.

Response: That’s correct. In the revised manuscript, we add a brief discussion on the model performance, bias and its spatial footprint: “*The overestimated historical and future enhancement of fire carbon emissions simulated by the default ESMs is mainly distributed across the historically sparsely vegetated regions (Extended Data Fig. 4), potentially as a result of unrealistic representation of dynamic vegetation processes⁴⁵. Furthermore, these ESMs display consistently strong linkage between the simulated historical and future fire carbon emissions (Extended Data Fig. 9), suggesting the primary need to improve historical wildfire simulation for better prediction of future wildfire evolution.*” (pages 14–15 lines 370–376)

L155. “more optimal” with “better”

Response: “More optimal” has been changed to “better” in the revised manuscript. (page 6 line 160)

L177-178. Was this calculated at pixel level or are R2 calculated for each ESM? If it is the first, acknowledging the regions with weaker performances will improve the understanding of the outputs.

Response: The spatial R² and correlation were calculated for each ESM and their multimodel mean, using all pixels on the globe. This sentence has been changed to “*e. Squared spatial correlation (R²) between the observed, decadal mean global fire carbon emission and the original (stars) and observation-constrained (boxplot) simulations from each of the 13 ESMs and their multimodel mean.*” (page 7 lines 184–186)

L284. Africa is not a county. Rephrase please.

Response: This sentence has been changed to “*Notably, although both the default and observation-constrained ensembles highlight African countries in their list of top 10 countries facing the greatest relative changes in socioeconomic risks from wildfires during the twenty-first century, the observation-constrained ensemble particularly tags west African countries, such as Niger and Sierra Leone, as the most vulnerable countries in all the three metrics of socioeconomic risks from future wildfires (Extended Data Table 3).*” (pages 11–12 lines 292–397)

L350-352. Actually the original approach shows a clear upward trend in emissions whereas the constrained model fluctuates over time (2040-2050) that can be consider a large relative change. The small changes are driven by the EMS that forecast a drop in that period but the differences disregarding those models wouldn’t that small.

Response: Please refer to our response to your major comment 1.

L371. Fuel, not biofuel.

Response: Instances of “biofuel” have been changed to “fuel” as suggested throughout the manuscript.

L382-383. Actually, most studies envisage that future extreme wildfires might be beyond suppression capability, regardless of the means available. See:

Moreira, F., Ascoli, D., Safford, H., Adams, M.A., Moreno, J.M., Pereira, J.M.C., Catry, F.X., Armesto, J., Bond, W., González, M.E., Curt, T., Koutsias, N., McCaw, L., Price, O., Pausas, J.G., Rigolot, E., Stephens, S., Tavsanoğlu, C., Vallejo, V.R., Wilgen, B.W. Van, Xanthopoulos, G., Fernandes, P.M., 2020. Wildfire management in Mediterranean-type regions: paradigm change needed. *Environ. Res. Lett.* 15, 11001. <https://doi.org/10.1088/1748-9326/ab541e>

Response: Thank you for the valuable thought and useful reference. We have elaborated the discussion on fire management in the revised manuscript: “*Over these fire-prone regions, projected elevation in mean fire states is likely accompanied with increased occurrence and intensity of extreme wildfire events that may grow beyond suppression capability, thereby requiring a paradigm shift in measuring the effectiveness of fire management policies*⁵⁷.” (page 16 lines 416–419)

L406-408. Can this connect with the more conservative estimates in fire-related emissions in the future??

Response: Since we do not have an additional, independent data set to test against, it is difficult to establish the connection between these observational data and the magnitude of fire-related emissions in the future.

L413. But this also increase the estimate in the original model, right?

Response: This sentence has been changed to “*Inclusion of such small fires in the observational data sets may result in an even greater magnitude of both historical and future global fire carbon emissions estimated by our observational constraint, likely as well as the default ensemble if tuned to match such observation.*” (page 17 lines 445–448)

L507-508. How is this similarity gauged?

Response: The whole paragraph is revised: “*The current spatial sample training approach establishes a history-future relationship for each pixel using the entire global sample. To minimize local prediction errors for a certain pixel, MLT search all pixels, regardless of their geographical location, to optimize the prediction model of future fires at the target pixel. In this way, a physically robust history-future relationship is established based on the global sample of locations, whereas influences of localized features, such as socioeconomic development, on wildfire trends are naturally damped in our approach (Extended Data Figs. 10 and 11). The reliability of MLT is degraded when the actual observational data space is insufficiently covered by the training (historical CMIP6 simulation) data space, namely the extrapolation uncertainty. Here, we further evaluate the data space of both observation and historical simulation of the climate and fire variables (Extended Data Fig. 14), and we find all these assessed variables are largely overlapped, indicating minimal extrapolation error involved in the current MLT application.*” (pages 20–21 lines 551–562)

L545. Rather increased observational data improving the spatial accuracy of the interpolation.

Response: There was an error in the caption of old Extended Data Fig. 7 (new Extended Data Fig. 8). The observational data sets are all at 0.25° resolution, regardless of the input model data resolution. This error was corrected: “*Because the MLT are trained using the global spatial sample, we expect the performance of MLT to be sensitive to the spatial resolution of the training data set. This assumption is tested by varying the interpolation grids (1°, 2.5°, 5°, and 10° latitude by longitude) of the ESMs and fitting MLT using this specific-resolution training data for the validation period (Extended Data Fig. 7). Observational data sets at 0.25° resolution are subsequently fed into the fitted MLT models, regardless of the input model data resolution. This sensitive test sheds light on the importance of spatial resolution to our observational constraining and thereby implies potential accuracy improvement of our MLT-based*

observation constraint with the development of higher-resolution ESMs.” (page 22 lines 601–609). Thus, this exercise suggests that increased model resolution tends to improve the spatial accuracy of the observational constraint.

L561. Interpolated or resampled?

Response: Thank you for the correction. In this analysis, we sum up the finer-resolution population, GDP, and agricultural area within 0.25° pixels. We believe “resampled” is the correct word. (page 23 line 623)

L588. I think average or median values plus standard deviation or IQR are more appropriate.

Response: We have elaborated the reason for choosing the highest importance score among the annual mean and 12 monthly mean values to represent the overall importance of a particular variable: “*For the atmospheric and terrestrial variables that include annual mean and monthly climatology as predictors, to account for the overall importance of a particular variable while considering the possible information overlapping contained in each month and annual mean, the importance of each variable is represented by the highest importance score among these 13 predictors (annual mean, January, February, ..., December).*” (page 24 lines 654–659)

Reviewer #2 (Remarks to the Author):

Yu et al NCC emergent constraints method for wildfire

The authors perform a variation of an emergent constraints analysis on earth system model projections of wildfire emissions under climate change. They use multiple fire relevant variables in combination with several machine learning techniques to constrain current and future estimates of the emissions from wildfire around the globe. They also estimate the exposure of population, GDP and agriculture to changes in emissions. Their technique does indeed adjust model outputs closer to observations, and results in a narrower, and generally lower estimate of the trajectory of future wildfire emissions, albeit one that is still rising steadily throughout the 21st century. Congratulations to the authors on an interesting and potentially useful study on an important topic. Improving projections of global wildfire activity is a very worthy task and the paper is generally well organised with good figures. Applying the emergent constraints paradigm to wildfire modelling is novel, as is the use of multiple variables and a machine learning approach. I have some high level concerns, as well as some more specific ones, which I detail below. In my opinion it is critical that an expert on EC be included among the reviewers. I am not an expert on emergent constraints, but it is not clear to me that the authors have avoided the pitfalls associated with this approach. Some of these are outlined in Hall et al. 2019, Williamson et al. 2021, Sanderson et al. 2021.

Response: Thank you for your positive feedback, valuable comments, and insightful references. In the revised manuscript, we have explicitly discussed (1) the EC pitfalls inducing limited reliability of traditional EC in the projection of fire carbon emissions, 2) the advances of our MLT-based EC framework over the traditional EC, and (3) the remaining uncertainties contained in most EC studies and incompletely addressed in this manuscript.

We have outlined the pitfalls associated with traditional EC that lead to the limited reliability of EC in the projection of fire carbon emissions: “*However, the traditional EC framework appears less reliable in the projection of fire carbon emissions (Extended Data Fig. 2), likely attributed to the following factors^{31,40,41}: (1) complex interactions among fire, climate, ecosystem, and socioeconomic variables, inadequately captured by the linear assumption adopted in the traditional EC framework; and (2) the relatively small*

collection of fire simulations and limited diversity in fire parameterizations for the available ESMs (Extended Data Table 1), providing insufficient sample for applying the traditional EC framework.” (page 13 lines 323–329)

We have clarified improvements of our framework over the traditional EC: *“Building on the EC concept, our MLT-based observational constraining framework advances further in several key perspectives: (1) This framework integrates the information from multitype observations of fire-relevant variables with the mechanistic history-future relationship encompassed in ESMs. (2) This framework relies on the power of MLT in capturing nonlinear and interactive relationships between the simulated historical joint states in fire-climate-ecosystem-socioeconomics and future wildfire activities. (3) By taking advantage of the complete spatial pattern provided by the relatively small collection of ESMs, this framework expands the sample size of the training data, thereby enhancing MLT model fitting efficiency and resolving the desired spatial pattern of future wildfire regimes.”* (page 13 lines 329–338). These methodological advances lead to the improved spatial accuracy of the MLT-based observational constraint over both the original ensemble and the traditional EC (Fig. 1).

We have discussed the uncertainties in the current framework that are shared by most EC studies and not completely avoided in the current study: *“Several uncertainties and limitations of the current study, as shared by most EC applications⁴¹, are noted here. First, the uncertainty in observational data may evolve into the current observational constraint. Although we analyze a spectrum of data sources for most climatic and ecosystem variables, the single data set used for lightning and socioeconomic variables, as well as the deteriorated reliability of reanalysis-based wind and specific humidity over observation-sparse regions⁶¹, likely leads to a weakened constraint gained from these variables. In addition, the currently examined fire carbon emission data sets were derived from relatively coarse-resolution satellite measurements of burned area (about 500 m resolution), which may miss nearly half of the burned area and associated carbon emissions in Africa as detected by higher-resolution satellite measurements (about 20 m resolution) in a given year⁶². Inclusion of such small fires in the observational data sets may result in an even greater magnitude of both historical and future global fire carbon emissions estimated by our observational constraint, likely as well as the default ensemble if tuned to match such observation. Second, the inconsistency between observed quantities and model-simulated variables limits further strengthening of our observational constraint. For example, the above ground biomass, as provided by most ESMs, more directly captures the amount of fuel than the combination of LAI, temperature, and precipitation, as used in our current analytical framework. Yet, a lack of long-term, reliable observational record of above ground biomass prohibits the direct use of such key driving variable in the current analysis. Third, the incomplete independence of the analyzed ESMs (e.g., CLM as the terrestrial component of several models) reduces the multimodel spread of the original ensemble, causing higher dependency of our MLT-based EC results on these shared modeling components as well. Moreover, as much as our observation-constrained ensemble provides bias-corrected historical and likely future fire carbon emissions, our approach does not directly account for more complex feedback from socioeconomic development to wildfire regimes beyond the prescribed parameterization of socioeconomic drivers of fire carbon emission involved in the currently examined ESMs. Future model development on socioeconomic-wildfire interactions, such as anthropogenic ignition, urbanization, prescribed burning, and anthropogenic suppression on natural ignitions, will enhance our confidence in predicting future socioeconomic risks from wildfires^{11,63,64}. Finally, the current MLT-based observation-constraint framework does not directly account for potential tipping points in fire regime evolution^{59,65} or certain threshold in fuel moisture content below which more intense fire behavior may occur²². The applicability of our framework to these extreme fire regimes needs further investigation.”* (pages 17–18 lines 436–468)

o Strength of the statistical relationship. As far as I can tell Supp Fig 2 is the only place that the actual performance of the model is displayed. While it is a significant improvement on the ‘default’ EC option

there are still large areas of fire-prone land with no significant correlation. This includes quite large parts of Australia, South America, Africa, North America and Europe, many of which are discussed at length in the manuscript. Can the authors clarify the amount of flammable land/burned area/emissions that their model is reliable for? Is it valid to apply the model in areas with no significant relationship? And does it matter that the model reproduces mean global emissions accurately in spite of these large areas without correlation? Could errors be cancelling each other out? Given the importance of this step, it is not clear to me that the authors have established the relationship significantly. They could perhaps provide some information to evaluate the machine learning approaches they have taken, either individually or in aggregate. No doubt an expert on EC could shed light on many of these issues.

Response: Extended Data Fig. 2 shows the limitations of traditional EC instead of the currently developed MLT-based framework. The unsatisfactory performance of traditional EC over Australia, South America, Africa, North America, and Europe motivates the development of the current MLT-based EC approach.

The performance of the MLT-based EC has been elaborated in the revised manuscript in terms of the MLT algorithms, or in-sample statistical strength: “*The prediction model is fitted for each MLT using the training data set that targets each future decade, with parameters optimized for the minimum RMSE via 10-fold cross-validation—in other words, using a randomly chosen nine-tenth of the entire spatial sample ($n = 10,193$) for model fitting and the remaining one-tenth of the entire spatial sample ($n = 1,132$) for validation, and repeating the process 10 times.... The cross-validation R^2 s exceed 0.8 ($n = 1,132$) for all optimized MLT and all future periods.*” (page 21 lines 570–583). Since we used the global spatial sample to train the MLTs, it is not straightforward to identify regions where in-sample statistical relationship is strong or weak.

In terms of actual performance of MLT-based EC, or out-of-sample statistical strength, Fig. 1 and associated text demonstrates the accuracy of this approach using multimodel ensemble and individual models during the validation period. Specifically, Fig. 1c shows the spatial distribution of statistical strength of the current approach, as described in the revised manuscript: “*The observation-constrained product substantially reduces the overestimation of fire carbon emissions over sparsely vegetated regions (mainly for EC-Earth3 models, Extended Data Fig. 1), tropical rainforests (mainly for EC-Earth3 models), northern boreal regions (mainly for MRI-ESM2.0), and densely populated regions in North America and Europe (mainly for CNRM-ESM2.1 and MPI-ESM1.2-LR), as well as the underestimation of fire carbon emissions over the savannahs in Africa from most analyzed ESMs (Fig. 1a–c). Relatively large error between the observation-constrained and observed fire carbon emissions remains in the present fire-prone regions (e.g., tropical and subtropical Africa, subtropical South America, and southeast Asia) (Fig. 1c).*” (page 6 lines 146–154)

o Insufficient sample size – is the use of individual pixels enough to overcome this?

Response: In the revised manuscript, we have clearly stated the sample size of our data pool: “*Corresponding to the spatial resolution of the observational products of fire carbon emission, all model outputs are bilinearly interpolated to a $0.25^\circ \times 0.25^\circ$ grid, resulting in a spatial sample of 11,325 points per model for the training.*” (page 20 lines 538–540) This sample is much larger than the traditional EC in the current study (13 models with 38 ensemble members). With this relatively large sample, the insufficient sample size issue faced by the traditional EC is overcome here.

o Model independence – multiple models share the same land use and fire parametrisations

Response: This model independence uncertainty is discussed in the revised manuscript: “*Third, the incomplete independence of the analyzed ESMs (e.g., CLM as the terrestrial component of several*

models) reduces the multimodel spread of the original ensemble, causing higher dependency of our MLT-based EC results on these shared modeling components as well.” (page 17 lines 454–457)

o Overlooking the potential for tipping points – these are widely recognised to be a risk for some fire regimes (Lenton et al. 2008)

Response: Thank you for pointing this out. We note this as a limitation of our study: *“Finally, the current MLT-based observation-constraint framework does not directly account for potential tipping points in fire regime evolution^{59,65} or certain threshold in fuel moisture content below which more intense fire behavior may occur²². The applicability of our framework to these extreme fire regimes needs further investigation.”* (page 17–18 lines 464–468)

o The mechanisms underpinning the relationship. I applaud the authors for including multiple relevant variables, but the use of machine learning largely obscures any mechanisms at play (the variable importance plots in the Supp Info notwithstanding). The authors describe this as process-based but I’m not sure this qualifies, as there is no explicit process. The authors do not provide any information on how they selected those variables (L518-520), or whether a different set might have performed better. Given the centrality of fire to this study, it would be good to see this addressed.

Response: Throughout the manuscript, we have clarified that “process-based” mainly means the use of Earth system models, for example, *“Process-based Earth system approaches, such as the use of Earth system models (ESMs), have the potential to account for many human-vegetation-fire-climate interactions and are thus suggested as a practical way to predict future changes of wildfire and associated socioeconomic exposure (e.g., population, gross domestic product [GDP], and agricultural area).”* (page 3 lines 51–54)

The selection of driving variables has been justified in the revised manuscript: *“Observed, historical environmental (e.g., fire carbon emission, leaf area index [LAI], soil moisture, temperature, precipitation, wind, relative humidity, flash rate, and orography) and socioeconomic (e.g., land use and population) variables (Extended Data Table 2 and reference therein) are subsequently fed into the trained MLT models, resulting in a multimodel, multi-data set ensemble of observation-constrained projections of future global distribution of fire carbon emissions for each decade. These driving variables of wildfires are selected so that their nonlinear combinations as determined by MLT reflect the fuel abundance⁸ (LAI, temperature, precipitation), fuel moisture³⁷ (soil moisture, relative humidity, precipitation, temperature), fire spread conditions³⁸ (wind and orography), and ignition sources³⁹ (flash rate, land use and population).”* (page 5 lines 128–137)

o Quality of observations. It is worth acknowledging limitations in this data. Wind speed in particular suffers from a lack of long term high quality observational datasets against which to judge the accuracy of reanalysis values. Relative humidity, flash rate and soil moisture are probably also a fair way behind variables like temperature and rainfall in terms of their accurate simulation. See L409

Response: Thank you for this comment. In the revised manuscript, we have added one new set of global observation-based soil moisture products (Supplemental Data Table 2); also, we have elaborated more on the observational data uncertainty: *“Although we analyze a spectrum of data sources for most climatic and ecosystem variables, the single data set used for lightning and socioeconomic variables, as well as the deteriorated reliability of reanalysis-based wind and specific humidity over observation-sparse regions⁶¹, likely leads to a weakened constraint gained from these variables.”* (page 17 lines 438–441)

o Can you clarify that you have avoided the possibility of strong but “overconfident” constraints (Sanderson et al. 2021)

Response: Based on the performance of our MLT-based observation-constraint during the historical validation period, we are confident that this framework effectively reduces model biases in the near future (i.e., using observations during 1997–2006 to constrain projections during 2007–2016), while this framework’s reliability in the longer term partially relies on the accuracy of physical processes contained in the ESMs and remains to be tested. We have clarified this point in the revised manuscript: *“Benefiting from the inclusion of the complete spatial sample, this observational constraint leads to a consistent and substantial error reduction in simulated global wildfire distribution (Fig. 1), demonstrating the robustness of our analytical framework even with just 13 ESM ensembles. Indeed, our MLT-based framework also shows satisfactory efficiency in error reduction with only 6 CMIP6 ESMs that simulate burned area fractions (Extended Data Fig. 7). Although the current MLT-based EC framework improves the spatial accuracy of original ESM-simulated fire carbon emissions during the historical validation period, the performance of our framework in the future decades partially relies on the accuracy of ESMs’ physical processes (e.g., complex responses of fire regimes to various natural and anthropogenic forcings) and must be further evaluated.”* (pages 13–14 lines 338–347)

o Can you give an indication of reasonable performance of EC and its basis? Presumably the process by definition improves accuracy, so what is the best way to judge its effectiveness?

Response: We have used historical data to test the out-of-sample performance of the MLT-based EC, and relevant results are shown in Fig. 1 and corresponding text (pages 6–7 lines 143–188).

- There are a number of simple errors in language and referencing which can easily be addressed, but somewhat undermine confidence in the rest of the manuscript e.g.

o It is risks from / posed by wildfire, not risks to wildfire

Response: Throughout the manuscript, it has been changed to “risks from wildfires.”

o Some references are listed twice

Response: Thanks! All double-listed references have been corrected.

o L91 bracket in wrong spot

Response: The bracket has been removed from this sentence.

o Abstract: Forkel et al. and Li et al. are good references but do not go to the claim that it is a lack of observational constraints that limit wildfire projection credibility. It is arguably a lack of process-based understanding that limits confidence on ESM wildfire simulation (Boer et al. 2021)

Response: The references in the Abstract have been removed, according to the format requirement of the journal.

o Language is a little too strong e.g. drastically, extremely disastrous, enormous, huge, remarkably, vast etc

Response: Throughout the revised manuscript, we have revised overly strong statements.

o Use of Wikipedia as a reference (I am a big fan of Wikipedia but don’t think it is appropriate as a source)

Response: Corresponding to the availability of reference from scientific journals, this statement has been changed to “*During the 2019–2020 Australian bushfire season, a series of major wildfires burned more than 81,000 km², costing more than 110 billion 2020 USD, and killing at least 28 people⁴.*” (page 2 lines 38–40)

o The reference to Abatzoglou et al (#7) is about fire weather, not fire behaviour

Response: The reference has been replaced with Brown, T., Leach, S., Wachter, B. & Gardunio, B. The Northern California 2018 extreme fire season. *Bull. Am. Meteorol. Soc.* **101**, S1–S4 (2020).

o Similarly, references #11 and #12 are about projections, whereas your claim is that changes have already occurred

Response: Thanks! These references have been replaced with the following:
Pechony, O. & Shindell, D. T. Driving forces of global wildfires over the past millennium and the forthcoming century. *Proc. Natl. Acad. Sci. U. S. A.* **107**, 19167–19170 (2010).
Marlon, J. R. *et al.* Climate and human influences on global biomass burning over the past two millennia. *Nat. Geosci.* **1**, 697–702 (2008).
Andela, N. *et al.* A human-driven decline in global burned area. *Science (80-.)*. **356**, 1356–1362 (2017).

o This may be a language issue, but refs #3 and #4 do not talk about feedbacks. Likewise #13 does not say that weaknesses in feedback representation lead to biases, it is more of a snap evaluation than a thorough one.

Response: In the first sentence, the reference has been replaced with: Harris, R. M. B., Remenyi, T. A., Williamson, G. J., Bindoff, N. L. & Bowman, D. M. J. S. Climate–vegetation–fire interactions and feedbacks: trivial detail or major barrier to projecting the future of the Earth system? *Wiley Interdiscip. Rev. Clim. Chang.* **7**, 910–931 (2016). The second sentence has been changed to “*Such uncertainties potentially lead to biases in the simulated historical fire carbon emission¹³ by ESMs participating in the latest Coupled Model Intercomparison Projection phase 6 (CMIP6)¹⁴ (Extended Data Fig. 1), casting doubt on the credibility of the projected wildfire evolution from the default models.*” (page 3 lines 57–60)

o #8 does not seem to actually discuss “complex, nonlinear integration of meteorological, ecological, and socioeconomic states”

Response: The reference has been replaced with the review paper on vegetation fire in the Anthropocene by Bowman et al. 2020.

o #43 would be better off being a general paper on prescribed burning rather than a projection of prescribed burning conditions e.g. Russell-Smith et al 2020.

Response: Thanks! That reference has been replaced with Russell-Smith et al. 2020 as suggested.

o #45 is about smoke not fire danger

Response: The phrase has been changed to “*real-time monitoring of smoke spread⁵³.*” (page 16 line 410)

o L320 should this be Supp Fig 2?

Response: Thanks for the correction!

o L343 you cannot say that you have made an accurate assessment of future risk. That remains to be seen

Response: Sure. This sentence has been changed to “*Such detailed structure of the target variable as projected by our MLT-based observational constraining framework facilitates accurate assessment of socioeconomic risks in the historical validation period (Fig. 3) and potentially improved future projections, leading to strategic implications for local and regional stakeholders.*” (page 14 lines 350–353)

o L367 onwards. I would be a bit more cautious in my language here, as you are speculating about the drivers of your results rather than actually testing them.

Response: To highlight our results, this sentence has been changed to “*The observation-constrained projection of pan-tropical enhancement in wildfire activities is likely affected by the changes in soil moisture and relative humidity (Extended Data Fig. 10), consistent with previous conclusions regarding accelerated drying over the tropics under climate change⁴⁶ and increased occurrence of severe tropical droughts^{47–49}.*” (page 15 lines 380–384)

o #48 lists the wrong authors

Response: We have corrected this reference and double-checked all references.

Other comments

- L73-78 can you rephrase to make this clearer? I’m not sure what you mean by sections or multisector states, nor how the listed references relate to this statement.

Response: Thank you for raising this concern. This sentence has been changed to “*Although current ESMs only include incomplete and highly parameterized driving processes for fire (Extended Data Table 1), the involved physics-based coupling among fire, climate, ecosystem, and human activities across different scales (e.g., consistent mechanistic relationships between variability in fire-relevant variables, such as air temperature, precipitation, and vegetation coverage, and persistent sensitivity of these climate or ecosystem variables to external anthropogenic forcings) sets the basis for linking future fires with historical states of these components in the ESMs^{23,24}.*” (page 3 lines 65–71)

- L126 A good citation is Jain et al. 2020

Response: Thanks for the useful reference. It is now included in the revised manuscript. (page 5 line 132)

- If possible it would be good to show a map of actual emissions (rather than % change) in the Supp Info (equivalent to Fig 2b,c,d)

Response: In Extended Data Fig. 5., we have shown the actual emission trend ($\text{kg m}^{-2} \text{yr}^{-1} \text{dec}^{-1}$) and the projected emission during the 2090s ($\text{kg m}^{-2} \text{yr}^{-1}$), estimated by both the original and observation-constrained ensembles.

- Likewise can the authors please show model bias e.g. Fig 1, Supp Fig 6 – don’t just show model and observations, but show the difference between the two

Response: In the updated Fig. 1 and Extended Data Fig. 7 (the old Extended Data Fig. 6), we have replaced the mean fire carbon emission maps with the bias maps.

- It is important to acknowledge that SSP-5 is not business as usual, but a (fortunately) less likely higher emissions scenario (Hausfather & Peters 2020). Ideally the authors would include a more realistic scenario as well, but at the least they could acknowledge this.

Response: Thank you for the good suggestion. To address the dependence of fire projection on emission scenario, we have applied our framework to 9 CMIP6 ESMs that provide SSP2-45 simulations (Extended Data Fig. 12) and have compared the results with the analysis conducted for SSP5-85 with the same set of models (Extended Data Fig. 13). The results and implications have been discussed in the revised manuscript: *“The projected fire regimes and their socioeconomic risks depend on the projected socioeconomic pathway. The currently examined SSP5-85 reflects a high-emission scenario⁵⁸, whereas a lower-emission scenario, SSP2-45, suggests a generally milder increase in global fire carbon emission, for both the original and observation-constrained ensembles (Extended Data Fig 12). In the northern subtropical and mid-to-high latitudes, while the default ensemble indicates a spatially homogeneous but slightly weaker increase in fire carbon emission in SSP2-45, compared with that estimated for SSP5-85 from the same set of ESMs (Extended Data Fig. 13), the observation-constrained ensemble indicates opposite sign of changes in the Appalachian Mountains of the United States in SSP2-45 and SSP5-85. Greater differences in the projected fire carbon emission between SSP5-85 and SSP2-45, in terms of both sign and magnitude of changes, are seen over the northern subtropics, tropics, and Southern Hemisphere. Such complicated dependency of future projection of fire regimes on socioeconomic pathways is likely attributed to the nonlinear interaction among fire, climate, vegetation, and human activity, as well as potential occurrence of tipping points in ecology and/or climate evolution^{59,60}. ”* (page 16 lines 321–434)

- It’s great that the authors include information about the fire model within ESMs in Supp Table 1, but these descriptions don’t mean much to me or I suspect most readers. Can the authors expand this to a short para, or in some other way clarify the general nature of these fire modules and how they differ?

Response: In the revised manuscript, we have added the following description of the ESMs and their fire models: *“These ESMs provide coupled carbon-ecosystem-climate simulations with a wide range of processes and parameterizations included¹⁴. Their terrestrial components typically contain fire models with process-based and/or data-based parameterization for various landscapes, accounting for effects of changes in both land surface meteorological states, vegetation-soil conditions and human activities on fire regimes (Extended Data Table 1 and reference therein).”* (page 5 lines 113–118)

- The country based approach is definitely valuable (Fig 3) as is the Congo and Amazon zoom in (Supp Fig 9). However, I think you will find better relationships and more meaningful drivers if you focus on dominant fire types e.g. forest fire, grass fire, peats etc.

Response: Nice suggestion! The projected fire carbon emission trends and their drivers are added in Extended Data Fig. 11 along with the Congo and Amazon (Extended Data Fig.10). The results are discussed in the revised manuscript: *“In particular, the observation-constrained ensemble projects increased wildfire activity over the Amazonian and Congo Basins, in contrast to the default simulation for Congo and to a larger extent for Amazon (Fig. 2b, c). The observation-constrained projection of pan-tropical enhancement in wildfire activities is likely affected by the changes in soil moisture and relative humidity (Extended Data Fig. 10), consistent with previous conclusions regarding accelerated drying over the tropics under climate change⁴⁶ and increased occurrence of severe tropical droughts⁴⁷⁻⁴⁹. Such apparent association between future drying and elevated fire carbon emission is also identified over other forest, grassland, and cropland, as estimated by the observation-constraint (Extended Data Fig. 11). In the Congo basin, the projected elevation in the amount of fuel⁵⁰, partially reflected by the positive contribution of leaf area index trends to the future increased fire carbon emissions as indicated by the observational constraints (Extended Data Fig. 10b), further supports a more flammable future. The leading role of fuel abundance in future fire regimes also appears in other forest, shrubland, savannahs,*

and cropland (Extended Data Fig. 11). Because of the global, spatial sampling approach (see Methods), our constraining approach results in a much weaker contribution of projected local socioeconomic development (e.g., population density and land use) to the projected trend in fire carbon emissions than the default ensemble, for all major land cover types (Extended Data Fig. 11). Although the parameterized anthropogenic source and suppression of wildfires in ESMs reflects valuable efforts to represent socioeconomic influence on wildfire regimes, their accuracy and applicability to future scenarios remain to be rigorously evaluated. In this perspective, our MLT-based observation-constrained ensemble raises an alternative scenario of future evolution of fires in the Congo region—with relative weak anthropogenic suppression and/or more anthropogenic ignitions than that estimated by CMIP6 ESMs.” (page 15 lines 378–400)

References

- Boer et al 2021 A Hydroclimatic Model for the Distribution of Fire on Earth
<https://doi.org/10.1088/2515-7620/abec1f>
- Hall et al. 2019 Progressing emergent constraints on future climate change
<https://doi.org/10.1038/s41558-019-0436-6>
- Hausfather & Peters 2020 Emissions – the ‘business as usual’ story is misleading.
<https://doi.org/10.1038/d41586-020-00177-3>
- Jain et al. 2020 A review of machine learning applications in wildfire science and management.
<http://dx.doi.org/10.1139/er-2020-0019>
- Lenton TM, Held H, Kriegler E, Hall JW, Lucht W, Rahmstorf S and Schellnhuber HJ 2008 Tipping elements in the Earth’s climate system Proc. Natl. Acad. Sci. USA 105 1786–93
- Russel-Smith et al 2020 Adaptive prescribed burning in Australia for the early 21st Century – context, status, challenges <https://doi.org/10.1071/WF20027>
- Sanderson et al. 2021 The potential for structural errors in emergent constraints
<https://doi.org/10.5194/esd-12-899-2021>
- Williamson et al. 2021 Emergent constraints on climate sensitivities
<https://doi.org/10.1103/RevModPhys.93.025004>

Reviewer #3 (Remarks to the Author):

This manuscript considers the use of machine learning techniques (MLTs) to better represent the nonlinear relationships between fire occurrence and various environmental factors at global scales, and to then use this to predict carbon release and socioeconomic impacts. Fire has traditionally been poorly represented in Earth system/global climate models and so studies like this one, which aim to improve the representation of fire are vitally important in providing an accurate picture of global change. The use of MLTs to better express the nonlinear relationships between interacting aspects of complex systems had success in other contexts, and so while their use is not new, the application to global fire occurrence and carbon release is particularly innovative. As such, the manuscript addresses a significant issue, and I am sure it will attract considerable scientific interest.

While my overall opinion is that the manuscript is worthy of publication, I feel that the manuscript could be strengthened by addressing a few issues that are not adequately addressed in the current version. These are as follows:

1. The authors mention the 2019-20 Australian fires, which are notable for the massive amount of carbon they released. Much of this occurred in connection with numerous episodes of extremely intense fire behaviour. These episodes have been linked to particular environmental factors, primarily rugged terrain, forest fuels and critically low fuel moisture content, and their interaction

(e.g., Sharples et al. 2016; Di Virgilio et al. 2019; Abram et al. 2021). While it could be argued that land-use accounts for forest fuels and that fuel moisture content is somewhat covered by temperature and relative humidity, the list of variables considered by the authors (Extended Data Table 2) doesn't include any terrain variables. I'm therefore left to wonder whether the author's analyses have identified all the relationships that are important to carbon release – particularly massive carbon release events.

Response: Thank you for your encouragement and your important suggestion on including terrain variables. Per simulation practices of CMIP6 models, we have included orography (mean elevation in a given pixel) as a predictor for constraining fire carbon emissions. The approach is introduced in the revised manuscript: *“Observed, historical environmental (e.g., fire carbon emission, leaf area index [LAI], soil moisture, temperature, precipitation, wind, relative humidity, flash rate, and orography) and socioeconomic (e.g., land use and population) variables (Extended Data Table 2 and reference therein) are subsequently fed into the trained MLT models, resulting in a multimodel, multi-data set ensemble of observation-constrained projections of future global distribution of fire carbon emissions for each decade. These driving variables of wildfires are selected so that their nonlinear combinations as determined by MLT reflect the fuel abundance⁸ (LAI, temperature, precipitation), fuel moisture³⁷ (soil moisture, relative humidity, precipitation, temperature), fire spread conditions³⁸ (wind and orography), and ignition sources³⁹ (flash rate, land use and population).”* (page 5 lines 128–137). All relevant figures and results have been updated. However, we notice the validation and projection results are not drastically changed after adding orography, likely because orography information is already contained in historical fire carbon emissions to some degree.

Thank you for suggesting adding the key references on driving processes of intensive fire behavior. We have included a statement of such processes in the revised manuscript: *“However, the linkages between fire weather and wildfire activity are greatly affected by other factors, including terrain, fuel abundance, fuel moisture content, source of ignition, and their interactions^{19–22}.”* (page 3 lines 63–65)

2. How does Leaf Area Index (LAI) account for the extensive regions of grasslands, savannah and other important vegetation types around the globe (e.g., xeric shrublands, spinifex, etc.)?? Moreover, how does satellite-derived LAI account for fuels in the surface and near-surface layers of forests, noting that it's the fuels within these layers (and their dryness) that have the greatest influence on fire occurrence and account for a considerable proportion of carbon release?

Response: Because of a lack of reliable, long-term observations on fuel abundance, our analysis uses LAI, precipitation, and temperature to reflect fuel abundance. Also, because of the lack of model outputs on fuel dryness, we have used temperature, precipitation, relative humidity, and soil moisture to reflect the fuel dryness. The reason why these driving variables were chosen is outlined in the revised manuscript: *“These driving variables of wildfires are selected so that their nonlinear combinations as determined by MLT reflect the fuel abundance⁸ (LAI, temperature, precipitation), fuel moisture³⁷ (soil moisture, relative humidity, precipitation, temperature), fire spread conditions³⁸ (wind and orography), and ignition sources³⁹ (flash rate, land use and population).”* (page 5 lines 134–137)

In the revised manuscript, we have also added the discussion on such uncertainty caused by observational data availability or modeling capability: *“Second, the inconsistency between observed quantities and model-simulated variables limits further strengthening of our observational constraint. For example, the above ground biomass, as provided by most ESMs,*

more directly captures the amount of fuel than the combination of LAI, temperature, and precipitation, as used in our current analytical framework. Yet, a lack of long-term, reliable observational record of above ground biomass prohibits the direct use of such key driving variable in the current analysis.” (page 17 lines 448–454)

3. Fuel moisture, which depends on air temperature and relative humidity, can have a significant effect on wildfire behaviour. For example, Abram et al. (2021) note that when fuel moisture content drops below certain thresholds, wildfire behaviour can be noticeably different. This is due to greater propensity for certain types of fire behaviour, such as spotting. My concern in this respect, is that these thresholds are not incorporated in the approach used by the authors, and so their analyses may be missing important dynamics that could occur more often in the future as conditions become warmer. In particular, the methods used may not account for more intense fire behaviour, which could release substantially more carbon. The authors should note this as a limitation of their study and revise their conclusions accordingly.

Response: Thank you for the good point. The use of MLTs addresses the nonlinear dependence of fuel moisture on soil moisture, air temperature, and relative humidity to some degree, as outlined in the revised manuscript: *“These driving variables of wildfires are selected so that their nonlinear combinations as determined by MLT reflect the fuel abundance⁸ (LAI, temperature, precipitation), fuel moisture³⁷ (soil moisture, relative humidity, precipitation, temperature), fire spread conditions³⁸ (wind and orography), and ignition sources³⁹ (flash rate, land use and population).”* (page 5 lines 134–137)

We have also noted this as a limitation of our study: *“Finally, the current MLT-based observation-constraint framework does not directly account for potential tipping points in fire regime evolution^{59,65} or certain threshold in fuel moisture content below which more intense fire behavior may occur²². The applicability of our framework to these extreme fire regimes needs further investigation.”* (pages 17–18 lines 464–468)

4. The release of carbon from global fire constitutes an important climate feedback. Was this accounted for in the study? That is, were the carbon release estimates found in the study used to further inform the climatic projections used in a coupled manner? Again, if not, this should be noted as a limitation of the study.

Response: Thank you for pointing this out. Our approach provides an offline fire projection constraint and does not account for climate or ecological feedbacks of fire carbon emission. This limitation has been addressed in the revised manuscript: *“Although the current approach does not account for climate or ecological feedbacks of global fire carbon emissions, dynamical coupling between observation-constrained fire carbon emissions and other components of the Earth system will likely result in a more reliable projection of all these components.”* (pages 14 lines 366–370)

5. I think that once these issues have been addressed the manuscript will be suitable for publication. However, I think doing so might constitute a major revision of the manuscript.

Response: Thank you again for your time and constructive suggestions.

References

Abram, N.J., Henley, B.J., Gupta, A.S., Lippmann, T.J., Clarke, H., Dowdy, A.J., Sharples, J.J., Nolan, R.H., Zhang, T., Wooster, M.J. and Wurtzel, J.B., 2021. Connections of climate change and variability to

large and extreme forest fires in southeast Australia. *Communications Earth & Environment*, 2(1), pp.1-17.

Di Virgilio, G., Evans, J.P., Blake, S.A., Armstrong, M., Dowdy, A.J., Sharples, J. and McRae, R., 2019. Climate change increases the potential for extreme wildfires. *Geophysical Research Letters*, 46(14), pp.8517-8526.

Sharples, J.J., Cary, G.J., Fox-Hughes, P., Mooney, S., Evans, J.P., Fletcher, M.S., Fromm, M., Grierson, P.F., McRae, R. and Baker, P., 2016. Natural hazards in Australia: extreme bushfire. *Climatic Change*, 139(1), pp.85-99.

Peer review comments, second round review–

Reviewer #2 (Remarks to the Author):

Yu et al have provided a comprehensive response to the reviewers' comments and the manuscript is significantly improved as a result. It was already strong but the authors have gone to great efforts to address comments and portray the work as clearly and honestly as possible, which I appreciate. I have a couple of minor comments which I believe can be addressed without further reviewer input. I recommend the paper be accepted. Congratulations to the authors on performing and reporting this work and all the best with your future endeavours.

1. L38-40 The authors cite Deb et al 2020, but if you read it it seems the primary source for the life loss and economic cost figures is Roach, J. (2020). Australia wildfire economic damages and losses to reach \$110 billion. In AccuWeather. State College, PA: AccuWeather. Retrieved from <https://www.accuweather.com/en/business/australia-wildfire-economic-damages-and-losses-to-reach-110-billion/657235>. I cannot find a reference to 81,000km² in Deb et al 2020. I suggest citing either Filkov et al 2020 (which includes figures of 33 lives lost, 190,000km² and multiple financial estimates) or Nolan et al. 2021 (which includes figures of 33 deaths, 72,000km² in the temperate south-east (the site of the most extraordinary fires) and some discussion of economic costs but no overall figure).

2. L130 I congratulate the authors for adding orography to their model, but I think reviewer 3 was referring to the ruggedness of terrain more than elevation as a driver of extreme fire behaviour and carbon emissions. It may be preferable to use a ruggedness metric that rather than mean elevation across a grid cell. However, i would be surprised if it made much of a difference given the scale you are working at. I think it suffices to acknowledge terrain as an uncertainty along with other factors, which you have already done.

References

Filkov et al 2020 <https://doi.org/10.1016/j.jnlssr.2020.06.009>

Nolan et al 2021 <https://doi.org/10.3390/fire4040097>

Reviewer #3 (Remarks to the Author):

I thank the authors for attending to the comments I provided. I am satisfied that they have adequately addressed my comments, but would like to note one small point of clarification regarding the use of orography as a predictor. Recent studies have found that when it comes to the effects of topographic factors on wildfires, it is not so much raw elevation that makes the difference, but derived quantities like slope and aspect, or more importantly for large carbon release events, terrain ruggedness. Terrain ruggedness can be defined as local relief (e.g. the difference between maximum and minimum elevation within a defined radius - typically about 1.5 km), but see Di Virgilio et al. (2019) and the references therein for further information. Hence, it is not surprising to me that just including orography did not alter the results very much.

I also appreciate the fact that representing topographic factors such as ruggedness is problematic in the context of ESMs, and I take the author's point about how topographic information may already be accounted for to a degree by historical carbon emissions. However, I think it would be worth mentioning the points about derived topographic factors I have made above, and perhaps highlight their incorporation (at appropriate scales) as one of the challenges that should be addressed in future studies of global wildfire/carbon regimes.

Other than that relatively minor issue, it is my opinion that the manuscript is now suitable for publication. I thank the authors for their interesting and important work.

REVIEWERS' COMMENTS

Reviewer #2 (Remarks to the Author):

Yu et al have provided a comprehensive response to the reviewers' comments and the manuscript is significantly improved as a result. It was already strong but the authors have gone to great efforts to address comments and portray the work as clearly and honestly as possible, which I appreciate. I have a couple of minor comments which I believe can be addressed without further reviewer input. I recommend the paper be accepted. Congratulations to the authors on performing and reporting this work and all the best with your future endeavours.

1. L38-40 The authors cite Deb et al 2020, but if you read it it seems the primary source for the life loss and economic cost figures is Roach, J. (2020). Australia wildfire economic damages and losses to reach \$110 billion. In AccuWeather. State College, PA: AccuWeather. Retrieved from <https://www.accuweather.com/en/business/australia-wildfire‐economic-damages- and- losses- to- reach- 110- billion/657235>. I cannot find a reference to 81,000km² in Deb et al 2020. I suggest citing either Filkov et al 2020 (which includes figures of 33 lives lost, 190,000km² and multiple financial estimates) or Nolan et al. 2021 (which includes figures of 33 deaths, 72,000km² in the temperate south-east (the site of the most extraordinary fires) and some discussion of economic costs but no overall figure).

Reply: Thank you for the useful references. We have corrected the numbers according to Filkov et al. (2020) and updated the reference (lines 36–39).

2. L130 I congratulate the authors for adding orography to their model, but I think reviewer 3 was referring to the ruggedness of terrain more than elevation as a driver of extreme fire behaviour and carbon emissions. It may be preferable to use a ruggedness metric that rather than mean elevation across a grid cell. However, i would be surprised if it made much of a difference given the scale you are working at. I think it suffices to acknowledge terrain as an uncertainty along with other factors, which you have already done.

Reply: Thank you for your comments about topographic influence on wildfire regimes. Due to the simulation practices and relatively coarse spatial resolution of CMIP6 models, it is not straightforward to include sub-grid terrain ruggedness as a predictor in our machine learning constraining model. In the revised manuscript, we explicitly discuss this uncertainty: “Second, the inconsistency between observed quantities and model-simulated or model-utilized variables limits further strengthening of our observational constraint... Another example involves the effects of sub-grid topography, such as slope, aspect, and terrain ruggedness at a typical scale of several kilometers, on wildfire spread and intensity⁶³. Future ESM development are encouraged to incorporate the sub-grid topographic factors to improve their representation of wildfire regimes and facilitate better observational constraint.” (lines 458–468)

References

Filkov et al 2020 <https://doi.org/10.1016/j.jnlssr.2020.06.009>

Nolan et al 2021 <https://doi.org/10.3390/fire4040097>

Reviewer #3 (Remarks to the Author):

I thank the authors for attending to the comments I provided. I am satisfied that they have adequately addressed my comments, but would like to note one small point of clarification regarding the use of orography as a predictor. Recent studies have found that when it comes to the effects of topographic factors on wildfires, it is not so much raw elevation that makes the difference, but derived quantities like slope and aspect, or more importantly for large carbon release events, terrain ruggedness. Terrain ruggedness can be defined as local relief (e.g. the difference between maximum and minimum elevation within a defined radius - typically about 1.5 km), but see Di Virgilio et al. (2019) and the references therein for further information. Hence, it is not surprising to me that just including orography did not alter the results very much.

I also appreciate the fact that representing topographic factors such as ruggedness is problematic in the context of ESMs, and I take the author's point about how topographic information may already be accounted for to a degree by historical carbon emissions. However, I think it would be worth mentioning the points about derived topographic factors I have made above, and perhaps highlight their incorporation (at appropriate scales) as one of the challenges that should be addressed in future studies of global wildfire/carbon regimes.

Other than that relatively minor issue, it is my opinion that the manuscript is now suitable for publication. I thank the authors for their interesting and important work.

Reply: Thank you for your suggestions about topographic influence on wildfire regimes. In the revised manuscript, we explicitly discuss this point: “Second, the inconsistency between observed quantities and model-simulated or model-utilized variables limits further strengthening of our observational constraint... Another example involves the effects of sub-grid topography, such as slope, aspect, and terrain ruggedness at a typical scale of several kilometers, on wildfire spread and intensity⁶³. Future ESM development are encouraged to incorporate the sub-grid topographic factors to improve their representation of wildfire regimes and facilitate better observational constraint.” (lines 458–468)